# Structural basis for DNA damage-induced phosphoregulation of MDM2 RING domain

Helge M. Magnussen [1,2], Syed F. Ahmed [1], Gary. J. Sibbet[1], Ventzislava A. Hristova[3], Koji Nomura[1,2], Andreas K. Hock[1,7], Lewis J. Archibald[4], Andrew G. Jamieson[4], David Fushman[5], Karen H. Vousden[6], Allan M. Weissman[3] & Danny T. Huang [1,2]✉

Phosphorylation of MDM2 by ATM upon DNA damage is an important mechanism for deregulating MDM2, thereby leading to p53 activation. ATM phosphorylates multiple residues near the RING domain of MDM2, but the underlying molecular basis for deregulation remains elusive. Here we show that Ser429 phosphorylation selectively enhances the ubiquitin ligase activity of MDM2 homodimer but not MDM2-MDMX heterodimer. A crystal structure of phospho-Ser429 (pS429)-MDM2 bound to E2–ubiquitin reveals a unique $3_{10}$-helical feature present in MDM2 homodimer that allows pS429 to stabilize the closed E2–ubiquitin conformation and thereby enhancing ubiquitin transfer. In cells Ser429 phosphorylation increases MDM2 autoubiquitination and degradation upon DNA damage, whereas S429A substitution protects MDM2 from auto-degradation. Our results demonstrate that Ser429 phosphorylation serves as a switch to boost the activity of MDM2 homodimer and promote its self-destruction to enable rapid p53 stabilization and resolve a long-standing controversy surrounding MDM2 auto-degradation in response to DNA damage.

[1] Cancer Research UK Beatson Institute, Garscube Estate, Switchback Road, Glasgow G61 1BD, UK. [2] Institute of Cancer Sciences, University of Glasgow, Glasgow G61 1QH, UK. [3] Laboratory of Protein Dynamics and Signaling, Center for Cancer Research, National Cancer Institute, Frederick, MD 21702, USA. [4] School of Chemistry, University of Glasgow, Joseph Black Building, G12 8QQ Glasgow, UK. [5] Department of Chemistry and Biochemistry, Center for Biomolecular Structure and Organization, University of Maryland, College Park, MD 20742, USA. [6] The Francis Crick Institute, London NW1 1AT, UK. [7] Present address: AstraZeneca, AstraZeneca R&D, Innovative Medicines, Discovery Sciences, Darwin (Building 310), Cambridge Science Park, Milton Road, Cambridge CB4 0WG, UK. ✉email: d.huang@beatson.gla.ac.uk

Often referred to as the guardian of the genome, p53 is an important tumor suppressor protein in human cells[1]. Upon activation by cellular stress, p53 binds DNA and activates transcription of target genes that initiate cell cycle arrest or apoptosis to repair or destroy the stressed cells, respectively. Consequently, mutations that reduce the ability of p53 to bind DNA are highly oncogenic. In fact, more than half of human tumor cells carry a *TP53* mutation, and most of these are directly connected to a reduced ability of p53 to recognize DNA[2]. Basal levels of p53 are maintained under normal conditions and rapidly elevated upon various cellular stresses. When cellular homeostasis is attained, p53 levels are attenuated through the ubiquitin (Ub)-proteasome system[3]. Thus, precise regulation of p53 activity is a key requirement for healthy cell growth.

There is a large and not yet fully understood network of proteins that are associated with p53 regulation. The most prominent regulator of p53 is the Ub ligase (E3) MDM2, which binds to the transactivation domain of p53 through its N-terminal p53-binding domain, thereby inhibiting p53's transcriptional activity[4–7]. Furthermore, MDM2 catalyzes ubiquitination of p53, where monoubiquitination promotes nuclear export of p53 and polyubiquitination leads to proteasomal degradation[8–11]. The C-terminal RING domain of MDM2 is essential for ubiquitination and requires dimerization with either itself[12] or its catalytically inactive homolog MDMX[13] to form an active homodimer or heterodimer, respectively. The importance of the homodimer and heterodimer in E3 activity and p53 regulation are underscored by mouse studies in which deletion of either *Mdm2* or *MdmX* or knock-in of a catalytically inactive *Mdm2* mutant results in early embryonic lethality due to uncontrolled p53 activity that is rescued upon concomitant deletion of *Trp53*[14–19]. A delay in the embryonic lethality in *MdmX*-knockout mice indicates that the heterodimer plays a role at a different stage of development that cannot be fulfilled by the homodimer.

Under unstressed conditions, basal MDM2 levels ensure that p53 levels are kept low[20–22]. However, in the presence of cellular stress, p53's anti-tumorigenic function needs to be activated quickly. One key aspect of this activation process is the uncoupling of p53 from MDM2-mediated regulation via post-translational modifications. Upon DNA damage, p53 is phosphorylated within its transactivation domain, thereby reducing its binding affinity for MDM2[23,24]; however, this is not sufficient to fully block MDM2 activity toward p53[25–28], suggesting that additional regulatory mechanisms exist. Phosphorylation of MDMX upon DNA damage also contributes to p53 stabilization: phosphorylation of MDMX by c-Abl impairs p53 binding[29], and phosphorylation by Chk2 and ATM kinases promotes binding of MDMX to MDM2 and subsequent ubiquitination and degradation by MDM2[30–32]. MDM2 also becomes phosphorylated upon DNA damage. Phosphorylation of MDM2's Y394 by c-Abl impairs the inhibitory effect of MDM2 on p53[33], and phosphorylation by CK1 near the acidic domain facilitates ubiquitination of MDM2 by the SCF$^{\beta\text{-TRCP}}$ complex and its subsequent degradation[34].

ATM-mediated phosphorylation of MDM2 has been shown to precede p53 stabilization after DNA damage[35]. ATM phosphorylates MDM2 on six residues (S386, S395, S407, T419, S425, and S429) that are close to the C-terminal RING domain and far from the domains associated with p53 binding[36]. Expression of MDM2 having phosphomimetics in place of all six residues impedes oligomerization, resulting in reduced p53 ubiquitination; this effect is also observed in native MDM2 upon ATM activation[36,37]. Moreover, phosphorylated MDM2 has a reduced half-life that may be a result of enhanced autoubiquitination[38] or increased susceptibility to ubiquitination by SCF$^{\text{FBXO31}}$ complex[39], thus leading to its degradation and p53 stabilization. MDM2 auto-degradation remains controversial as this phenotype

was attributed to a loss of epitope recognition by MDM2 antibodies after phosphorylation[40]. Studies on MDM2's S395 showed that phosphorylation or aspartic acid substitution leads to p53 stabilization, whereas alanine substitution stabilizes MDM2 and causes defects in DNA damage-induced p53 stabilization[41–43], highlighting the importance of S395 phosphorylation in MDM2 destabilization. While these cellular correlations are informative and intriguing, it remains unclear how phosphorylation promotes MDM2 degradation and whether, with six phosphorylation sites present, multiple mechanisms may come into play.

We have previously elucidated the crystal structure of the MDM2-MDMX RING heterodimer bound to E2 covalently linked to Ub (E2–Ub; the conjugated Ub is referred to as donor Ub)[44]. In this complex, MDM2's S429 is in close proximity to the donor Ub, suggesting that phosphorylation at this residue might affect E2–Ub recruitment. Here, we show that S429 phosphorylation selectively enhances MDM2 homodimer E3 activity, but has no effect on MDM2-MDMX heterodimer E3 activity. To understand the molecular basis for this selectivity, we determine the crystal structures of native and pS429-MDM2 RING homodimer in complex with E2–Ub. Comparison of the structures together with biochemical analyses reveals the basis for how pS429 stabilizes E2–Ub to enhance E3 activity. We show that pS429-mediated enhanced activity promotes MDM2-catalyzed autoubiquitination in cells after DNA damage, thus resulting in its rapid degradation. These results provide insights into phosphoregulation of MDM2.

## Results

**Ser429 phosphorylation boosts the activity of MDM2 homodimer**. ATM phosphorylates MDM2 at a region near the C-terminal RING domain. Given the proximity of phosphorylation sites and the RING domain, we asked whether aspartic acid substitutions of these residues influence E2–Ub binding. We generated an MDM2 RING domain construct (350–C) containing six aspartic acid substitutions (S386D, S395D, S407D, T419D, S425D, S429D; MDM2-6D) and measured its binding affinity for UbcH5B–Ub by using surface plasmon resonance (SPR) analyses. Like in our previous study, UbcH5B–Ub was generated by mutating UbcH5B's S22 to arginine and the catalytic C85 to lysine to eliminate the contribution of backside Ub binding and to form a stable isopeptide bond with Ub's C terminus to mimic the thioester linkage, respectively[45]. We found that MDM2-6D exhibited an ~2.3-fold enhancement in binding affinity for UbcH5B–Ub compared to wild-type MDM2 (hereafter MDM2-WT indicates wild-type (WT) version of different MDM2 fragments) (Table 1; Supplementary Fig. 1). Single aspartic acid substituted variants showed that S429D substitution was the main contributor to the binding enhancement (Table 1). A shorter MDM2 fragment containing residues 419–C displayed a similar affinity for UbcH5B–Ub as MDM2-350–C (Table 1), indicating that residues 350–418 are not involved in UbcH5B–Ub binding. Consistent with this, introduction of an S429D or S429E substitution in the MDM2-419–C fragment also improved the affinity for UbcH5B–Ub by ~2-fold compared to MDM2-WT (Table 1), suggesting that a negative charge at S429 was responsible for the binding enhancement.

To verify that phosphorylated S429 also improves UbcH5B–Ub binding, we utilized the translational insertion of O-phosphoserine system[46] to generate MDM2-419–C-pS429. Phosphorylation was validated by Western blot using a customized antibody specific for MDM2-pS429 (Supplementary Fig. 2a). Mass spectrometry analysis confirmed the presence of pS429, but unphosphorylated peptide was also detected (Supplementary Fig. 2b). Nonetheless, MDM2-pS429 exhibited an ~2.7-fold enhancement in binding

affinity for UbcH5B–Ub compared to MDM2-WT. This binding enhancement is similar to that observed for the S429D and S429E phosphomimetic mutants. We reasoned that the true binding enhancement upon phosphorylation might be more dramatic as

our sample contained unphosphorylated MDM2. Strikingly, when S429E substitution was introduced into the MDM2-MDMX RING domain heterodimer, it had minimal effect on the binding affinity for UbcH5B–Ub compared to wild-type MDM2-MDMX (Table 1).

To assess whether the enhanced UbcH5B–Ub binding affinity improves E3 activity, we performed in vitro autoubiquitination assays using glutathione *S*-transferase (GST)-tagged MDM2-419–C or cleaved MDM2-419–C. MDM2-S429E displayed an increased autoubiquitination activity compared to MDM2-WT (Fig. 1a, b and Supplementary Fig. 3a–d). In contrast, replacing MDM2-S429E in the MDM2-MDMX heterodimer had no effect on activity, consistent with our binding analyses (Fig. 1c, d and Supplementary Fig. 3e). By following the autoubiquitination reaction of the MDM2 homodimer over time, we found that the S429E substitution improved the rate by ~1.6-fold as compared to WT (Fig. 1e, f and Supplementary Fig. 3f). Moreover, under our reaction conditions the ubiquitination patterns appeared to be similar — the ubiquitinated products of the MDM2-WT reaction after 120 s were comparable to the ubiquitinated products of the MDM2-S429E reaction after 60–80 s (Fig. 1e). Thus, the phosphomimetic substitution of S429 affects the rate of catalysis, but not the nature of the ubiquitination pattern.

**Structure of MDM2 RING domain homodimer bound to UbcH5B–Ub.** The effect of the MDM2 S429 substitution on

**Table 1 Dissociation constants $K_d$ of MDM2 homodimer and MDM2-MDMX heterodimer variants for UbcH5B–Ub.**

| Ligand | $K_d$ (μM) |
|---|---|
| MDM2-350-C | 15.7 ± 0.6 |
| MDM2-350-C-6D | 6.7 ± 0.2 |
| MDM2-350-C-S386D | 13.7 ± 0.3 |
| MDM2-350-C-S395D | 13.7 ± 0.3 |
| MDM2-350-C-S407D | 13.7 ± 0.3 |
| MDM2-350-C-T419D | 13.0 ± 0.5 |
| MDM2-350-C-S425D | 13.4 ± 0.7 |
| MDM2-350-C-S429D | 7.8 ± 0.3 |
| MDM2-419-C | 14.0 ± 0.3 |
| MDM2-419-C-S429D | 7.1 ± 0.7 |
| MDM2-419-C-S429E | 6.4 ± 0.4 |
| MDM2-419-C-pS429 | 5.1 ± 0.4 |
| cat MDM2-422-C | 16.8 ± 0.3 |
| cat MDM2-422-C-S429E | 6.8 ± 0.4 |
| MDM2-350-C/MDMX-428-C | 36.7 ± 2.1 |
| MDM2-350-C-6D/MDMX-428-C | 29.2 ± 0.6 |
| MDM2-419-C/MDMX-418-C | 23.4 ± 0.8 |
| MDM2-419-C-S429E/MDMX-418-C | 17.8 ± 1.4 |

Standard error of mean values are indicated. Number of replicates, representative sensorgrams, and binding curves are shown in Supplementary Fig. 1.

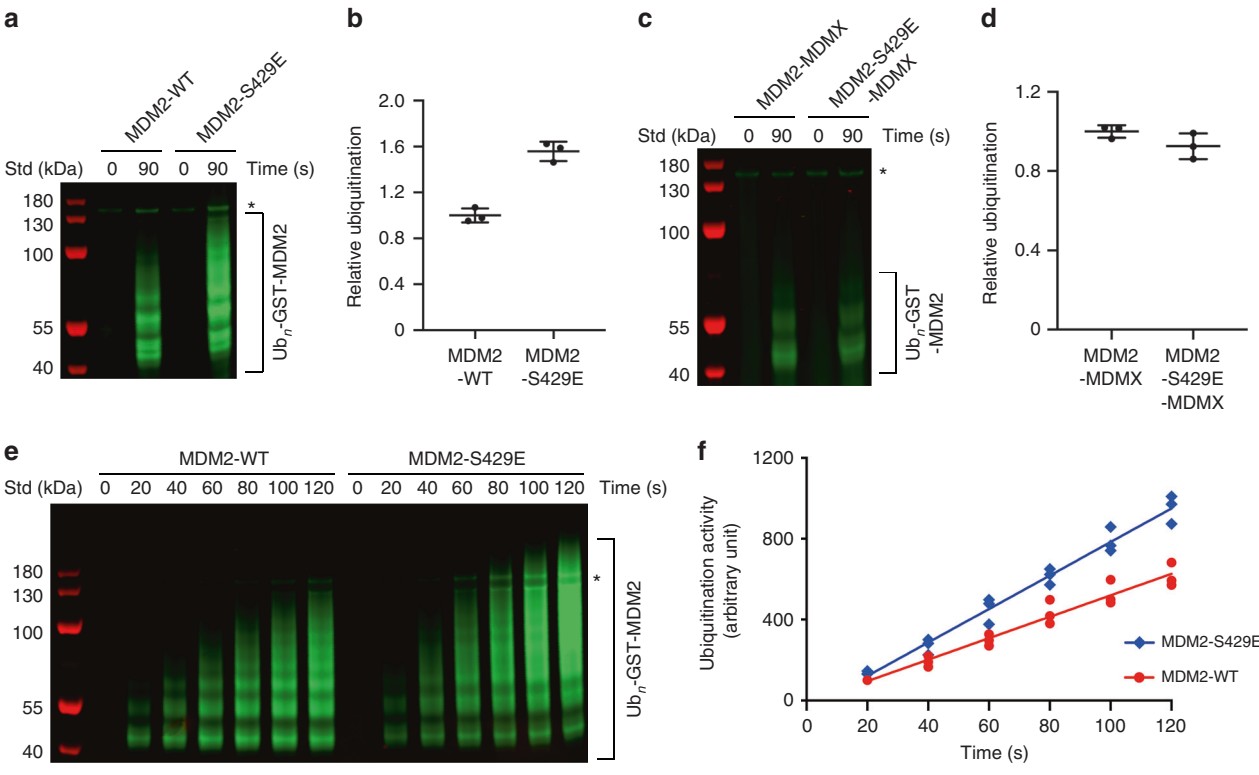

**Fig. 1 Phosphomimetic substitution at S429 enhances the E3 activity of the MDM2 homodimer. a** Reduced SDS-PAGE showing autoubiquitination reactions catalyzed by GST-MDM2-419–C and its S429E substitution using fluorescently labeled Ub and visualized with an Odyssey CLx Imaging System. **b** Plot of relative ubiquitination activity corresponding to **a**. **c** Reduced SDS-PAGE showing autoubiquitination reactions catalyzed by GST-MDM2-419–C-His-MDMX-418–C and its MDM2-S429E substitution using fluorescently labeled Ub and visualized with an Odyssey CLx Imaging System. **d** Plot of relative ubiquitination activity corresponding to **c**. For **b**, **d**, data are presented as mean value ± SD from three independent experiments (n = 3). **e** Reduced SDS-PAGE showing autoubiquitination reactions catalyzed by GST-MDM2-419–C and its S429E substitution over the indicated times using fluorescently labeled Ub and visualized with an Odyssey CLx Imaging System. **f** A plot showing the rates of ubiquitination catalyzed by the forms of MDM2 assessed in **e**. The line represents the regression line from three independent experiments (n = 3). Asterisks in **a**, **c**, and **e** indicate non-reducible E1–Ub product. Uncropped gel images and InstantBlue-stained gels are shown in Supplementary Fig. 3.

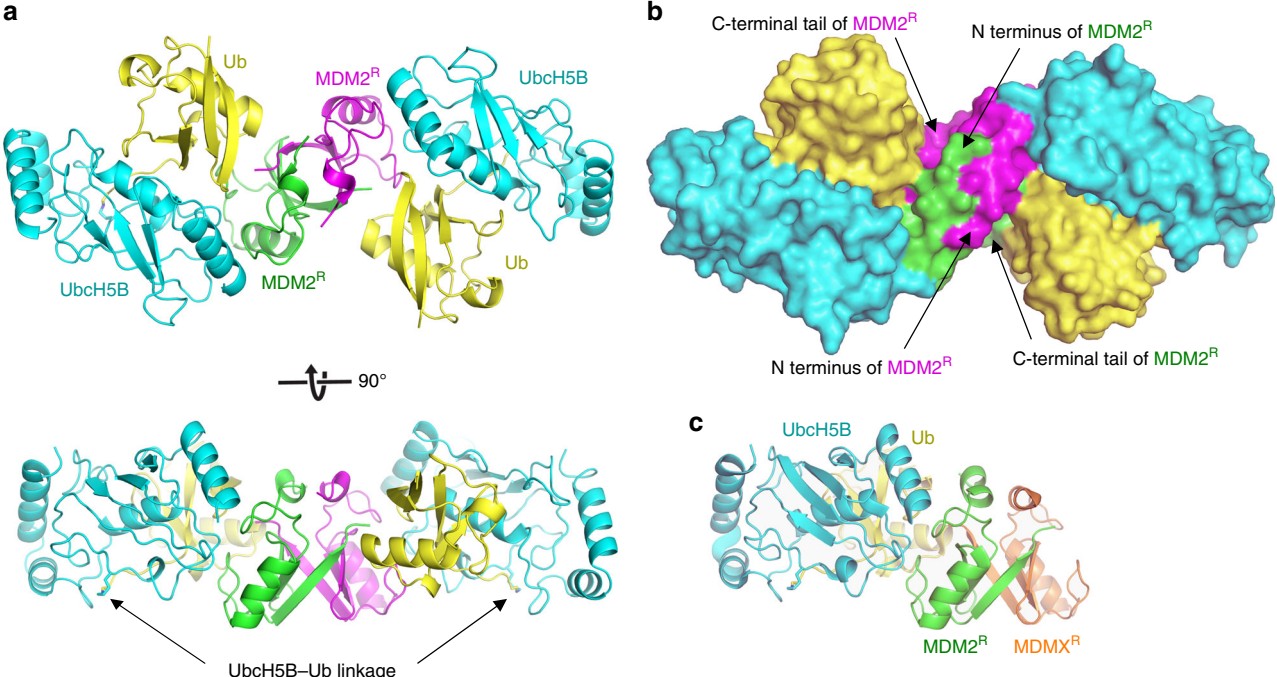

**Fig. 2 Crystal structure of human MDM2 RING domain homodimer bound to UbcH5B–Ub. a** Cartoon representation of the complex. One MDM2$^R$ is colored green and the other is in magenta. UbcH5B is in cyan and Ub is in yellow. UbcH5B–Ub linkage is indicated. Top and bottom panels are related by 90° rotation about the x-axis. **b** Surface representation of the complex, colored and oriented as in **a** (top panel). The N-terminal region preceding the RING domain and the C-terminal tail of MDM2 are indicated. **c** A cartoon representation of the structure of MDM2-MDMX RING domain heterodimer bound to UbcH5B–Ub (PDB ID: 5MNJ) oriented in the same view as in **a** (bottom panel). MDM2$^R$, UbcH5B, and Ub are colored as in **a** and MDMX RING domain (MDMX$^R$) is in orange.

UbcH5B–Ub binding suggests that the MDM2 RING domain homodimer and the MDM2-MDMX RING domain heterodimer use distinguishable mechanisms to recruit UbcH5B–Ub. We have recently determined the structure of the MDM2-MDMX RING domain heterodimer bound to UbcH5B–Ub and proposed a model for how the MDM2 homodimer recruits E2–Ub[44]. However, this model does not explain how pS429 exclusively affects MDM2 homodimer. To better understand the E2–Ub recruitment mechanism of the homodimer, we determined the structure of the human MDM2 RING homodimer (419–C; referred to as MDM2$^R$ and hereafter superscript R indicates the RING domain) in complex with UbcH5B–Ub to 1.41 Å (Fig. 2a, b and Table 2). The asymmetric unit contains two molecules of MDM2$^R$ that form a dimer, where each MDM2$^R$ binds a UbcH5B–Ub molecule in a similar manner (root mean square (r.m.s.) deviation of 0.16 Å for Cα atoms between the two monomeric portions of MDM2$^R$-UbcH5B–Ub complex). This is in contrast to the MDM2-MDMX heterodimer, which only binds one molecule of UbcH5B–Ub (Fig. 2c)[44].

MDM2$^R$ binds UbcH5B and Ub and arranges UbcH5B–Ub into a closed conformation similar to that observed in the MDM2-MDMX-UbcH5B–Ub structure[44] (r.m.s. deviation of 0.33 Å for Cα atoms of the MDM2$^R$-UbcH5B–Ub complex) and other RING E3-E2–Ub complexes[47,48]. This conformation is primarily stabilized by MDM2$^R$-UbcH5B, MDM2$^R$-Ub, and Ub-UbcH5B interactions (Fig. 3a, b) that are common to both MDM2 homodimer and MDM2-MDMX heterodimer. We have previously highlighted the importance of these interactions by showing that disruption of these interactions reduced the E3 activity of both the MDM2 homodimer and the MDM2-MDMX heterodimer[44]. Additionally, the dimeric arrangement enables the C-terminal tail from the second RING domain protomer to stabilize Ub in the closed conformation *in trans* (Fig. 2a, b). In the

MDM2 homodimer, the last three residues of MDM2 (Y489, F490, and P491) from the second MDM2$^R$ protomer are buried within the MDM2$^R$-Ub interface and pack against the Gly35 surface of Ub *in trans* (Fig. 3c), whereas in the MDM2-MDMX heterodimer, this is fulfilled by the last three residues of MDMX (F488, I489, and A490)[44]. This tail–Ub interaction is observed in other dimeric RING E3-E2–Ub complexes[47,48], thus explaining the importance of dimerization. Consistent with our structure, alterations in MDM2's C-terminal tail sequence were previously shown to hinder the activity of the homodimer[44,49–51].

A major structural difference between the dimers lies in MDM2's residues 430–436, which precede the RING domain (Fig. 3d, e). These residues are not involved in crystal packing contacts (Supplementary Fig. 4a) and are cradled by the N-terminal region and C-terminal tail of the second MDM2$^R$ protomer such that residues 432–436 adopt 3$_{10}$-helices in each MDM2$^R$ protomer. The helices stabilize each other via hydrophobic interactions involving A434 and I435 (Figs. 2b and 3d). L430 and P431 adopt an extended configuration and are stabilized by hydrophobic interactions involving I435, P437, P445, and F490 from the second MDM2$^R$ protomer (Fig. 3d). In the MDM2-MDMX-UbcH5B–Ub complex structure, MDM2's residues 428–436 form a continuous α-helical structure, whereas MDMX's residues 428–436 adopt a 3$_{10}$-helical structure (Fig. 3e). While the crystal packing contacts could contribute to the structural configurations observed in the heterodimer (Supplementary Fig. 4b,c), it seems likely that the structural differences between the homodimer and the heterodimer are caused by sequence differences of MDM2$^R$ and MDMX within the N-terminal region (residues 430–436) and the C-terminal tail, which pack against the N-terminal region *in trans* (Figs. 2 and 3d, e). Consequently, the residues preceding E436 of MDM2$^R$ in the homodimer are shifted in comparison to the heterodimer, leaving

**Table 2 Data collection and refinement statistics.**

| | Human MDM2-419–C-UbcH5B–Ub | Cat MDM2-422–C-pS429-UbcH5B–Ub | Cat MDM2-422–C-S429E-UbcH5B–Ub | Cat MDM2-422–C-S429E |
|---|---|---|---|---|
| **Data collection** | | | | |
| Space group | $P6_1$ | $P1$ | $P2_1$ | $P2_1$ |
| Cell dimensions | | | | |
| $a, b, c$ (Å) | 129.7, 129.7, 70.5 | 54.6, 56.4, 60.7 | 56.5, 163.9, 70.6 | 29.2, 39.8, 104.4 |
| $\alpha, \beta, \gamma$ (°) | 90, 90, 120 | 66.44, 69.44, 89.1 | 90, 96.03, 90 | 90, 93.4, 90 |
| Resolution (Å) | 112-1.41 (1.43-1.41)[a] | 29.2-1.83 (1.88-1.83) | 70-2.18 (2.22-2.18) | 23.53-1.21 (1.24-1.21) |
| $R_{merge}$ (%) | 4.7 (66.4) | 4.0 (26.0) | 9.4 (71.1) | 10.0 (84.5) |
| $I/\sigma I$ | 19.2 (1.9) | 10.2 (1.4) | 9.4 (1.6) | 5.1 (1.1) |
| Completeness (%) | 100 (99.6) | 95.8 (82.3) | 98.8 (98.3) | 98.3 (98.2) |
| Redundancy | 8.9 (6.0) | 1.8 (1.8) | 3.4 (3.5) | 3.2 (3.1) |
| CC (1/2) | 1.0 (0.542) | 0.996 (0.539) | 0.994 (0.56) | 0.991 (0.608) |
| Wilson $B$ (Å$^2$) | 17.89 | 30.93 | 31.0 | 10.43 |
| **Refinement** | | | | |
| Resolution (Å) | 112-1.41 | 29.2-1.83 | 70-2.18 | 23.53-1.21 |
| No. of reflections | 129,794 | 49,057 | 65,509 | 71,796 |
| $R_{work}/R_{free}$ | 0.139/0.173 | 0.211/0.242 | 0.166/0.227 | 0.171/0.212 |
| No. of atoms | | | | |
| Protein | 4,619 | 4,527 | 9,072 | 2,045 |
| Ligand/ion | 16 | 4 | 68 | 14 |
| Water | 546 | 180 | 420 | 243 |
| $B$ factors | | | | |
| Protein | 25.83 | 38.54 | 38.43 | 16.31 |
| Ligand/ion | 38.31 | 28.80 | 44.73 | 22.49 |
| Water | 39.75 | 37.52 | 36.94 | 26.88 |
| R.m.s. deviations | | | | |
| Bond lengths (Å) | 0.005 | 0.004 | 0.007 | 0.005 |
| Bond angles (°) | 0.743 | 1.271 | 0.857 | 0.705 |
| Ramachandran | | | | |
| Favored (%) | 97.7 | 96.3 | 96.9 | 95.4 |
| Outlier (%) | 0 | 0 | 0 | 0 |

[a]Values within the parentheses are for highest-resolution shell.

them in different local environments to initiate distinct UbcH5B–Ub contacts. In the homodimer, N433 forms hydrogen bonds with the backbone carbonyl atom of F490 and the C-terminal carboxylate of P491 in the second MDM2$^R$ protomer, and these residues in turn form hydrogen bonds with the K11 and T9 sidechains of Ub, respectively, to stabilize the closed Ub conformation (Fig. 3d). In contrast, the continuous α-helical structure of MDM2 in the heterodimer forms a complementary contact with Ub's K11 surface (Fig. 3e).

To assess the importance of the N-terminal region of MDM2$^R$ in the context of the homodimer, we introduced arginine substitutions to disrupt the 3$_{10}$-helical Ub-binding feature and performed autoubiquitination reactions to assess their effects. MDM2 L430R, N433R, A434R, I435R, and E436R variants had reduced activities as compared to WT (Fig. 3f, g; Supplementary Fig. 5a), suggesting that the observed structural features are important for stabilization of E2–Ub in the closed conformation. In our structure, there is no electron density for residues 419–429, indicating that this region is disordered. However, the differences in the structural features of residues 430–436 could account for the effect of S429 phosphorylation in the homodimer compared to the heterodimer.

**Phosphorylated Ser429 contacts the donor Ub.** To understand the molecular basis of S429 phosphorylation on MDM2 activity, we purified human MDM2$^R$-pS429 with the aim to crystallize it in complex with UbcH5B–Ub. Due to the low protein yield of the O-phosphoserine system, we were only able to screen around the crystallization condition of human MDM2$^R$-UbcH5B–Ub

complex and did not obtain crystals. The sequence of MDM2 RING domain (437–C) and S429 are well conserved among mammalian species (Supplementary Fig. 6). For example, the RING domains of human and *Felis catus* (cat) MDM2 are identical, whereas the adjacent N-terminal region differs by only four residues (Fig. 4a). To assess whether cat MDM2 is also regulated by S429 phosphorylation, we generated cat MDM2-422–C-S429E and performed autoubiquitination assays and UbcH5B–Ub binding analyses. We showed that the S429E substitution enhanced the activity (Fig. 4b, c and Supplementary Fig. 5b) and UbcH5B–Ub binding affinity (Table 1) compared to WT, suggesting that the pS429-mediated activity enhancement is conserved. We then purified cat MDM2-422–C-pS429 using the translational insertion of O-phosphoserine system[46], confirmed the phosphorylation status by Western blot (Supplementary Fig. 2a), and assembled the complex with UbcH5B–Ub.

Diffracting crystals were obtained from crystallization conditions similar to those employed for the human MDM2$^R$-UbcH5B–Ub complex and used to determine the structure to 1.83 Å (Fig. 4d and Table 2). The asymmetric unit contains one MDM2 homodimer bound to two UbcH5B–Ub molecules arranged in configuration similar to the one in the structure of the human MDM2-UbcH5B–Ub complex (r.m.s. deviation of 1.06 Å for all Cα atoms). For both MDM2 protomers, electron density is visible for residues 428–C. Because both S429 and pS429 were evident in our protein preparation, we reasoned that the abundance of pS429 could influence its occupancy. However, we cannot exclude the possibility that only the phosphorylated species of cat MDM2 crystallized in complex with UbcH5B–Ub. Electron density was observed for all atoms of pS429 at a level of

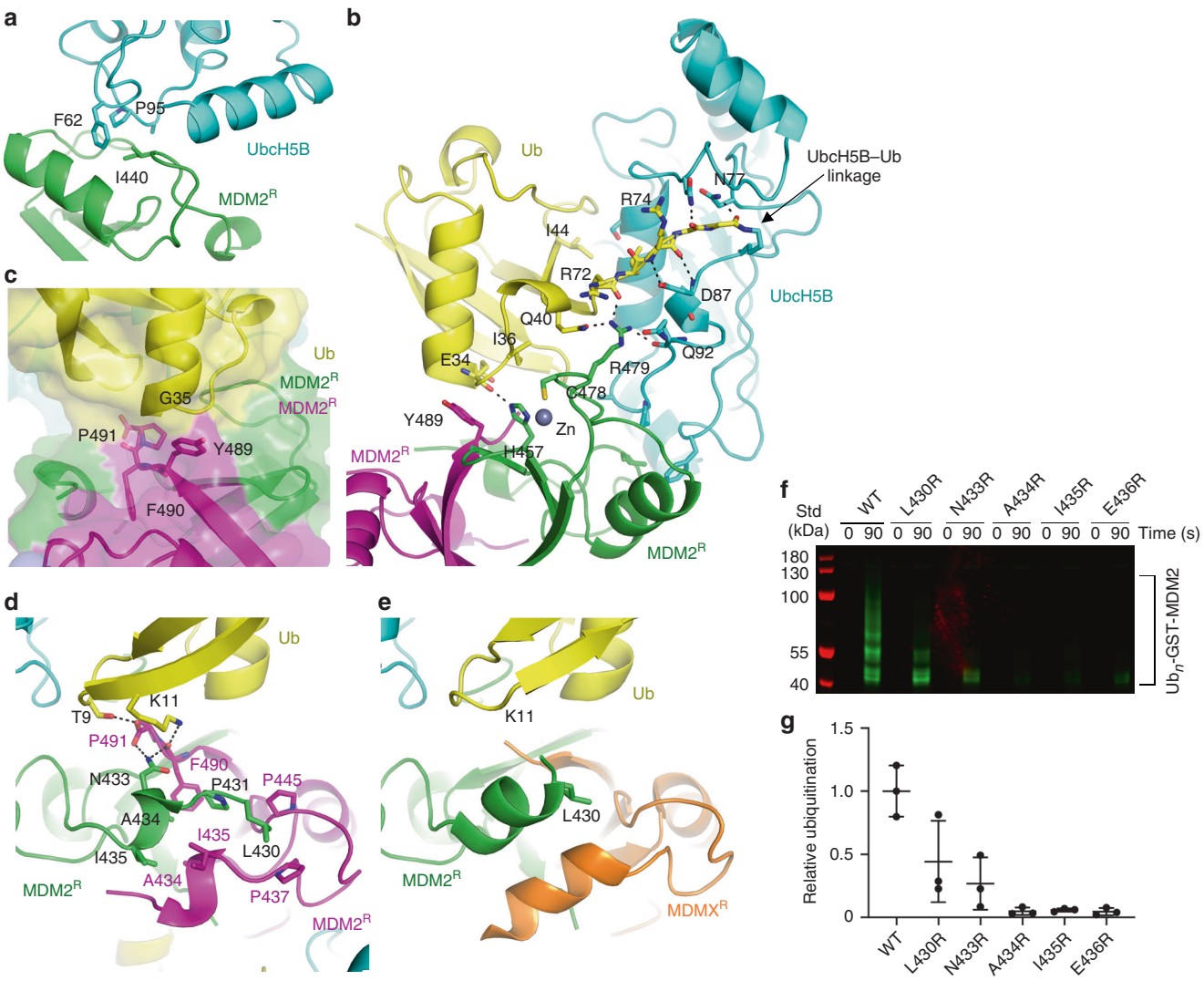

**Fig. 3 Importance of MDM2 residues in stabilizing the closed UbcH5B–Ub conformation. a** Close-up view of MDM2$^R$-UbcH5B interactions. **b** Close-up view of MDM2$^R$-Ub and Ub–UbcH5B interactions. **c** Close-up view of MDM2's C-terminal tail. A transparent surface representation is shown. **d** Close-up view of the N-terminal region preceding the MDM2 RING domain. **e** Close-up view of the N-terminal region preceding the RING domain in the structure of the MDM2-MDMX-UbcH5B–Ub complex (PDB ID: 5MNJ) shown in the same view as in **d**. **a**–**e** are colored as in Fig. 2. Key residues are shown as sticks. Carbon atoms are colored according to the parent subunit. Nitrogen, oxygen, and sulfur atoms are in blue, red, and gold, respectively. Zinc atoms are depicted as gray spheres. A dashed line indicates hydrogen bonds. **f** Reduced SDS-PAGE showing autoubiquitination reactions catalyzed by GST-MDM2-419–C and variants using fluorescently labeled Ub and visualized with an Odyssey CLx Imaging System. Uncropped gel images and InstantBlue-stained gels are shown in Supplementary Fig. 5a. **g** Plot of relative ubiquitination activity of MDM2 variants in **f**. Data are presented as mean value ± SD from three independent experiments ($n = 3$).

$1.0\sigma$ in the calculated polder map (Fig. 4e). Despite amino acid differences between human (L430 and L432) and cat (F430 and H432) MDM2 sequences, the structural features of the N-terminal region preceding the RING domain are conserved. Like in human MDM2, cat MDM2 residues 432–436 adopt a $3_{10}$-helical turn and F430 and P431 are stabilized by hydrophobic interactions involving I435, P437, P445, and F490 from the second MDM2 protomer (Fig. 4f). Thus, neither the sequence discrepancies nor phosphorylation of S429 introduced structural rearrangements.

In the cat MDM2 structure, the phosphate moiety of pS429 contacts Ub by forming hydrogen bonds directly with the ε-amino group of K33 and indirectly with the backbone amide of T14 via a water molecule (Fig. 4f). To further validate these interactions, we also determined the crystal structure of cat MDM2-422–C-S429E bound to UbcH5B–Ub (Table 2 and

Supplementary Fig. 7a). In this structure, the interactions of the sidechain of S429E in MDM2 resemble those of the pS429 phosphate moiety in the MDM2-pS429-UbcH5B–Ub complex: the carboxylate moiety of the S429E sidechain in MDM2 forms a hydrogen bond with the K33 sidechain of Ub and coordinates a water molecule that forms a hydrogen bond with the backbone amide of Ub's T14 (Fig. 4g, h). In both structures, crystal packing is facilitated by residues that precede the RING domain and that are unique to cat MDM2 (Supplementary Fig. 7b,c).

To validate the S429E–Ub interaction, we mutated Ub's K33 to methionine to disrupt the hydrogen bond and performed pulse-chased autoubiquitination assays in which UbcH5B was charged with Ub-K33M and then mixed with human MDM2-WT or the MDM2-S429E variant. Both MDM2 variants exhibited similar activity with Ub-K33M (Fig. 4i, j and Supplementary Fig. 5c), suggesting that the S429E-mediated enhanced activity with

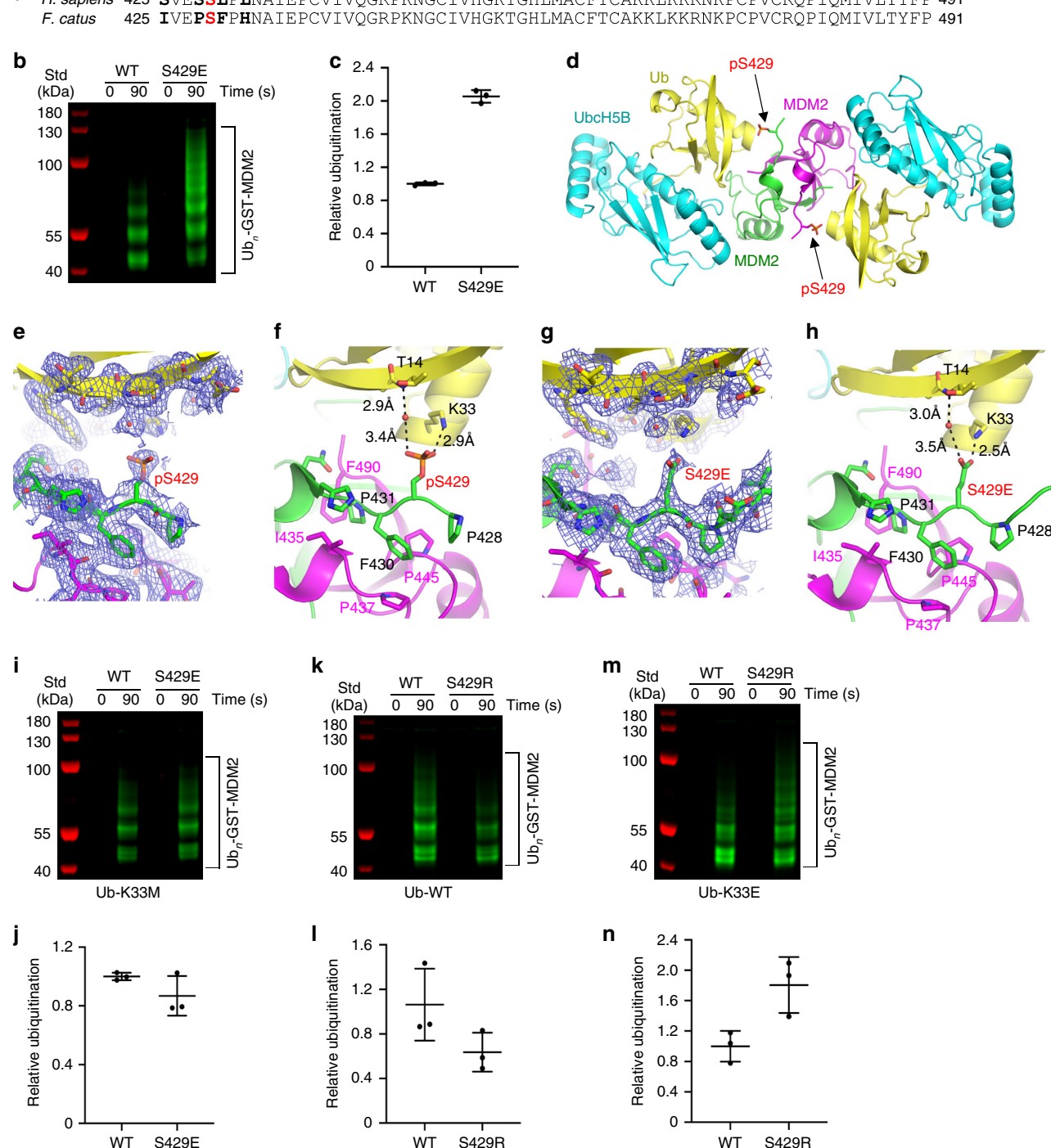

**Fig. 4 Molecular basis of S429 phosphorylation. a** Sequence alignment of MDM2 RING domain from human and cat. S429 is colored red. The four non-identical residues between human and cat are highlighted in bold. **b** Reduced SDS-PAGE showing autoubiquitination reactions catalyzed by GST-cat-MDM2-422–C and its S429E substitution using fluorescently labeled Ub and visualized with an Odyssey CLx Imaging System. **c** Plot of relative ubiquitination activity of cat MDM2 variants in **b**. **d** Cartoon representation of the structure of cat MDM2-422–C-pS429 bound to UbcH5B–Ub shown in the same colors and orientation as in Fig. 2a (top panel). pS429 is indicated. **e** Close-up view of pSer429 in **d** with polder density map (blue) contoured at 1σ. **f** Close-up view of pS429-Ub interactions in **d**. **g** Close-up view of S429E in cat MDM2-422–C-S429E-UbcH5B–Ub structure (Supplementary Fig. 7a) with polder density map (blue) contoured at 1σ. **h** Close-up view of S429E-Ub interactions as in **g**. For **e–h**, key residues are shown as sticks and colored as in Fig. 3. Phosphorus atoms are in orange. The water molecule is depicted as a red sphere. Hydrogen bonds are shown as dashed lines and the distances are indicated. **i**, **k**, **m** Reduced SDS-PAGE showing autoubiquitination reactions catalyzed by GST-MDM2-419–C variants using indicated fluorescently labeled Ub variants visualized with an Odyssey CLx Imaging System. **j**, **l**, **n** Plots showing the relative ubiquitination activity of MDM2 variants in **i**, **k**, **m**. In **c**, **j**, **l**, **n**, data are presented as mean value ± SD from three independent experiments (n = 3). All uncropped gel images and InstantBlue-stained gels are shown in Supplementary Fig. 5b–e.

Ub-WT (Fig. 1a, b) was abolished. To further establish the importance of the S429E-K33 charge–charge interaction, we swapped the charges on MDM2 and Ub by introducing S429R in MDM2 and K33E in Ub and performed pulsed-chased autoubiquitination assays. In the presence of Ub-WT, the MDM2-S429R variant exhibited reduced activity as compared to MDM2-WT (Fig. 4k, l and Supplementary Fig. 5d), suggesting that the Arg substitution might cause a charge repulsion with Ub's K33. In contrast, when Ub-K33E was used, MDM2-S429R displayed enhanced activity compared to MDM2-WT (Fig. 4m, n and Supplementary Fig. 5e), suggesting that the favorable charge interaction between these residues is responsible for the observed increased activity. Collectively, these results show that pS429 stabilizes the donor Ub in the closed conformation to enhance MDM2 activity.

**Importance of Ser429 location in the homodimer**. Our structures and biochemical data suggest that the location of pS429 or S429E with respect to the RING domain is important for optimal hydrogen bond formation with Ub's K33. In all the structures of MDM2 homodimer bound to UbcH5B–Ub presented here, the N-terminal region preceding the MDM2 RING domain adopts the same $3_{10}$-helical turn configuration. A prior solution structure of MDM2 RING domain homodimer determined by nuclear magnetic resonance showed that this region is flexible and adopts various conformations[12], suggesting that the $3_{10}$-helical turn configuration observed in our structures might be a product of UbcH5B–Ub binding. Since flexible or disordered regions usually lack electron density in crystal structures, we also crystallized and determined the structure of cat MDM2-422–C-S429E to 1.21 Å (Fig. 5a and Table 2). The asymmetric unit contains two dimers. The electron density for residues 429–C was visible and MDM2 adopts the same configuration as seen in the cat MDM2-422–C-S429E-UbcH5B–Ub structure (r.m.s. deviation of 0.30 Å for all Cα atoms). Despite contributing to crystallographic contacts, the region preceding the RING domain is able to adopt the same $3_{10}$-helical turn configuration independent of E2–Ub binding.

MDM2-MDMX RING domain heterodimer structures have previously been determined in the absence and presence of E2–Ub. In the structure of MDM2-MDMX-UbcH5B–Ub complex, the region preceding the RING domain of MDM2 adopts a continuous α-helical structure[44] (Fig. 5b, left panel). Superimposition of the MDM2-MDMX-UbcH5B–Ub complex structure onto the MDM2-pS429-UbcH5B–Ub complex structure shows that S429 of MDM2 in the heterodimer is displaced by ~10 Å (distance between the Cα atoms) as compared to pS429 of MDM2 in the homodimer (Fig. 5c). In comparison, in the structure of MDM2-MDMX alone (PDB ID: 2VJF)[13], residues 428–436 from MDM2 form a $3_{10}$-helical turn similar to the one formed by these residues in the homodimer (Fig. 5b, right panel). Superimposition of the MDM2-MDMX structure onto MDM2-pS429-UbcH5B–Ub shows that S429 of MDM2 in the heterodimer is ~3.5 Å away (distance between the Cα atoms) from pS429 of MDM2 in the homodimer and that its sidechain faces away from Ub's K33 (Fig. 5d). These structural comparisons show that the distance and trajectory of the S429 sidechain in the heterodimer preclude interactions with Ub's K33, thereby explaining why S429E substitution had no effect on the activity of the heterodimer. Closer inspection of homodimer and heterodimer structures reveals that variations in MDM2 and MDMX sequences in the region preceding the RING domain and in their C-terminal tail cause different structural configurations of MDM2's residues 429–431 in both dimers (Fig. 5e) such that only S429 in the homodimer can contact Ub upon phosphorylation.

To assess the importance of the position of pS429 in the MDM2 homodimer, we performed various autoubiquitination assays using the MDM2-S429E variant. First, we inserted an alanine residue between S429 and L430 in MDM2 to displace the position of S429E. Alanine insertion in MDM2-WT had a slight reduction in activity, whereas insertion in MDM2-S429E abolished S429E-mediated activity enhancement (Fig. 5f, g and Supplementary Fig. 5f). Second, we disrupted the $3_{10}$-helical turn configuration by introducing the aforementioned A434R substitution. A434R alone reduced the activity compared to MDM2-WT and was not enhanced by the additional incorporation of S429E substitution (Fig. 5h, i and Supplementary Fig. 5g). Third, we replaced MDM2's C-terminal residues with the MDMX sequence (F490I and P491A) to disrupt the packing surrounding S429. The MDM2-F490I-P491A mutant had reduced activity as compared to MDM2-WT and incorporation of S429E did not improve the activity (Fig.5j, k and Supplementary Fig. 5h). Collectively, these results underscore the importance of the structural features surrounding S429 in enabling precise positioning of pS429 for Ub interaction in the homodimer.

**Ser429 phosphorylation destabilizes MDM2 upon DNA damage**. Previously, S429D substitution in MDM2 was reported to stabilize p53 in H1299 cells[36]. We wanted to investigate whether this stabilization could be attributed to the enhanced MDM2E3 activity that we observed in vitro. We previously generated a U2OS cell line that lacks endogenous MDM2 and contains a doxycycline-inducible p53 short hairpin RNA (shRNA) (hereafter referred to as U2OS[mod] cells) to study the effects of MDM2 mutations in cells[44]. To investigate the relative stability of MDM2-WT and the phosphomimetics MDM2-S429D and MDM2-S429E, we transfected U2OS[mod] cells with the aforementioned GFP-MDM2 variants and followed their stability in a cycloheximide chase experiment. Both MDM2-S429D and MDM2-S429E were found to be less stable as compared to MDM2-WT (Fig. 6a). Consistent with this observation, reductions in the protein levels of MDM2-S429D and MDM2-S429E were evident in the total cell lysate as compared to MDM2-WT (Fig. 6b). To investigate whether the reduced stability of the MDM2 phosphomimetics correlates with enhanced autoubiquitination activity, MDM2-S429E was incorporated with a secondary site mutation I440K that blocks the RING domain from binding E2–Ub and has previously been shown to abolish MDM2's E3 function without influencing its ability to dimerize[44]. The MDM2-S429E-I440K variant was stabilized and did not get ubiquitinated (Fig. 6b), indicating that the observed ubiquitinated species were indeed products of autoubiquitination. Next, we investigated the effect of MDM2-S429 phosphomimetics on p53 stability by co-transfecting Myc-tagged p53 and GFP-MDM2 variants into unmodified U2OS cells. Cells expressing MDM2-S429D and MDM2-S429E displayed a slightly higher level of p53 as compared to cells expressing MDM2-WT (Fig. 6c). Correspondingly the p21 expression level was higher in cells expressing MDM2-S429 phosphomimetics. The ligase-dead MDM2-S429E-I440K caused p53 stabilization, but the p21 level was lower than that observed in the cells expressing empty vector (EV), suggesting that this MDM2 variant can still restrain p53 activity consistent with our prior observation[44]. Together, these results demonstrate that MDM2-S429 phosphomimetics exhibit enhanced activity in cells, but because the increased activity promotes its autoubiquitination and degradation, there is less MDM2 available to facilitate substrate degradation.

Since S429 is phosphorylated after DNA damage, we also performed cycloheximide chase experiments to compare the relative stability of MDM2-WT and MDM2-S429A in U2OS[mod]

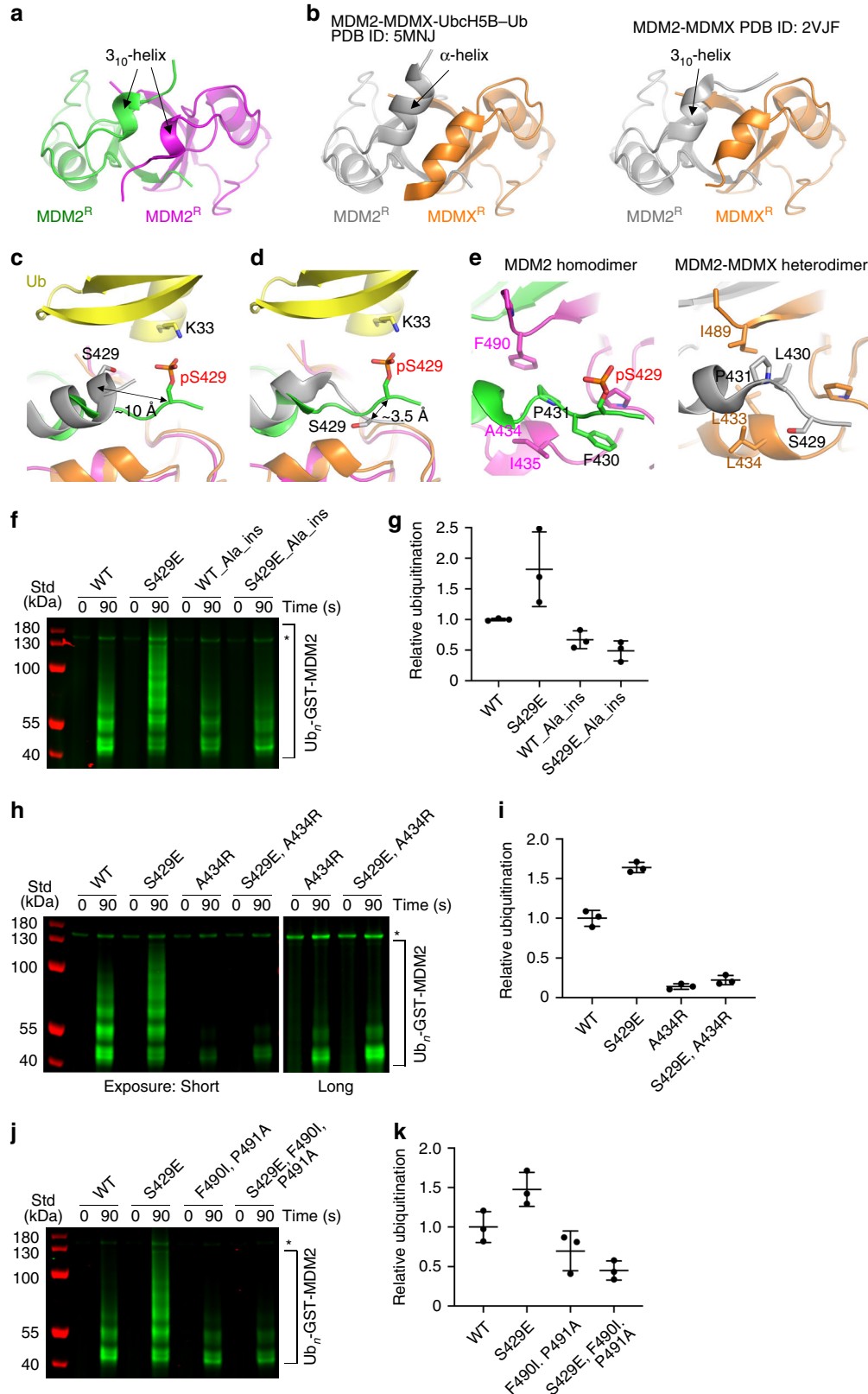

cells in the presence and absence of DNA damage-inducing etoposide treatment. We used GFP-tagged MDM2 and detected with anti-GFP antibody to avoid anti-MDM2 antibody epitope-masking effects by MDM2 phosphorylation after DNA damage[40]. There was no observable difference in the relative stability of the MDM2 variants in the absence of etoposide stress (Fig. 6d).

However, etoposide-induced DNA damage led to faster degradation of MDM2-WT, suggesting that the MDM2-S429A was partially protected from DNA damage-induced downregulation. Consistent with this, etoposide treatment led to an increase in MDM2-S429 phosphorylation, confirmed by Western blot using a customized MDM2-pS429 antibody (Fig. 6e). Trace amounts of

**Fig. 5 Importance of the structural features at the N terminus of the MDM2 RING domain homodimer. a** Cartoon representation of the structure of cat MDM2-422–C-S429E. **b** Cartoon representation of the MDM2-MDMX portion of MDM2-MDMX-UbcH5B–Ub complex structure (PDB ID: 5MNJ; left panel) and MDM2-MDMX RING domain structure (PDB ID: 2VJF; right panel) shown in the same view as in **a**. **c** Comparison of the location of pS429 in cat MDM2-422–C-pS429-UbcH5B–Ub structure and S429 in MDM2-MDMX-UbcH5B–Ub structure (PDB ID: 5MNJ). The MDM2-MDMX RING domain from the structure of MDM2-MDMX-UbcH5B–Ub complex was superimposed onto the structure of cat MDM2-422–C-pS429-UbcH5B–Ub complex. **d** Comparison of the location of pS429 in cat MDM2-422–C-pS429-UbcH5B–Ub structure and S429 in MDM2-MDMX RING domain structure (PDB ID: 2VJF). The MDM2-MDMX structure was superimposed onto the structure of cat MDM2-422–C-pS429-UbcH5B–Ub complex. **e** Close-up views of dimer interaction at the N terminus of MDM2 in the structures of cat MDM2-422–C-pS429-UbcH5B–Ub complex (left panel) and MDM2-MDMX RING domain (PDB ID: 2VJF; right panel). For **a**–**e**, all coloring is as described in Fig. 2, except MDM2$^R$ in the heterodimer is colored gray. Key residues in **c**–**e** are shown in sticks and atoms are colored as in Fig. 4f. Distances between the Cα atoms of pS429 and S429 are indicated in the homodimer in **c** and in the heterodimer in **d**. **f**, **h**, **j** Reduced SDS-PAGE showing autoubiquitination reactions catalyzed by GST-MDM2-419–C variants using fluorescently labeled Ub and visualized with an Odyssey CLx Imaging System. Asterisks indicate non-reducible E1–Ub product. In **f**, Ala_ins indicates insertion of an alanine between residues 429 and 430 in MDM2. Uncropped gel images and InstantBlue-stained gels are shown in Supplementary Fig. 5f–h. **g**, **i**, **k** Plots showing the relative ubiquitination activity of MDM2 variants in **f**, **h**, **j**, respectively. Data are presented as mean value ± SD from three independent experiments ($n = 3$).

---

MDM2-pS429 were detected in untreated cells, indicating that S429 phosphorylation might occur at a basal level even in the absence of DNA damage. Etoposide-treated cells also showed enhanced MDM2-WT autoubiquitination concomitant to it being highly phosphorylated at S429 under similar conditions (Fig. 6f). In contrast, differences in ubiquitination of MDM2-S429A with and without etoposide treatment were minimal (Fig. 6f). Collectively, these results suggest that under DNA damage-induced stress, MDM2 undergoes S429 phosphorylation, which enhances its autoubiquitination and subsequent degradation, and may eventually lead to p53 stabilization.

## Discussion

Our present work provides a mechanism by which pS429 enhances MDM2's E3 activity to promote its self-ubiquitination and rapid degradation upon DNA damage. Our structure of native MDM2-UbcH5B–Ub complex shows that the MDM2 homodimer is an active and competent E3 in the absence of pS429. The MDM2 homodimer utilizes the RING domain and the C-terminal tail to arrange UbcH5B–Ub into the closed cata-lytically active conformation via a mechanism similar to the MDM2-MDMX heterodimer and other dimeric RING E3s[44,47,48]. As the activity of RING E3s is governed by their ability to populate E2–Ub into the closed conformation[52], pS429 increases the activity by providing an additional interaction with the donor Ub to reinforce the closed E2–Ub conformation. Thus, S429 phosphorylation does not function as an off/on switch to robustly activate the E3 activity as observed in E3s such as CBL[53] and PARKIN[54]. Instead, it serves to mildly boost MDM2's activity. This mechanism might have evolved as a fine-tuning tool in mammals where S429 is mostly conserved (Supplementary Fig. 6). In other classes of animal, such as fish, this residue is not conserved despite high sequence identity RING domain residues, indicating that it is not essential for MDM2's E3 function.

ATM phosphorylates six residues (S386, S395, S407, T419, S425, and S429) adjacent to MDM2's RING domain upon DNA damage[36]. How do these phosphorylation sites regulate MDM2? Studies on MDM2's S395 showed that the S395D mutant exhibited similar stability as WT under unstressed conditions and was rapidly degraded like WT following DNA damage, whereas the S395A mutant was protected from degradation[42,43]. These findings suggest that phosphorylation of S395 together with other phosphorylation sites destabilizes MDM2. Our present study shows that S429 phosphorylation or phosphomimetic substitu-tion destabilizes MDM2 by increasing its ubiquitination activity in a manner that is dependent on MDM2's RING domain as the ligase-dead I440K substitution protects MDM2 from ubiquitina-tion. These results suggest that DNA-damage-induced S429

phosphorylation enhances MDM2's activity and accelerates MDM2 self-ubiquitination to promote its rapid degradation. Our data show that pS429 only exerts its effect in the context of the MDM2 homodimer. Thus, while ATM-mediated MDM2 phos-phorylation was reported to decrease MDM2 oligomerization[36], MDM2 must be able to dimerize for pS429 to promote its autoubiquitination. In previous studies, it was unclear whether MDM2 autoubiquitination caused a decrease in MDM2 protein levels after DNA damage as they relied on MDM2 antibodies that failed to recognize phosphorylated MDM2[40]. We circumvented this issue by using GFP-tagged MDM2 and detecting with anti-GFP antibody. Moreover, our structures and biochemical analyses provide strong evidence that DNA damage-induced S429 phos-phorylation enhances MDM2 autoubiquitination and degrada-tion. It is noteworthy that other E3s such as SCF$^{FBXO31}$ complex[39], p300/CBP-associated factor[55], anaphase-promoting complex/cyclosome[56], and SCF$^{β-TRCP}$ complex[34] have been reported to promote MDM2 ubiquitination and degradation after DNA damage. While it remains unclear how multiple E3s, including MDM2 itself, can act in a concerted manner or func-tion specifically under certain cellular contexts, the availability of multiple mechanisms ensures that MDM2 is rapidly eliminated to allow p53 stabilization.

We show that pS429 only enhances the activity of the MDM2 homodimer, but does not affect the activity of the MDM2-MDMX heterodimer. Comparison of the structures of the MDM2 homodimer and the MDM2-MDMX heterodimer bound to UbcH5B–Ub reveals a distinct structural feature in the homo-dimer that allows pS429 to interact with Ub. Our findings suggest that ATM-mediated MDM2 phosphorylation has distinct effects on the MDM2 homodimer and MDM2-MDMX heterodimer. Prior studies showed that MDMX is degraded by MDM2 after DNA damage[57,58]. This process is likely achieved by the hetero-dimer as MDMX with a C-terminal tag, which blocks E2–Ub binding and abolishes the heterodimer's activity[44], is resistant to ubiquitination and degradation by MDM2[57]. It remains unclear whether ATM phosphorylates MDM2 in the context of the MDM2-MDMX heterodimer after DNA damage, but a study showed that unlike MDM2, the rate of DNA damage-induced MDMX degradation is not appreciably affected by the MDM2 S395A mutant[43], suggesting that ATM-mediated MDM2 phos-phorylation does not regulate heterodimer stability. Future stu-dies are required to address the complexity of phosphoregulation of MDM2 in the context of homodimer and heterodimer.

Our study shows that MDM2 S429 phosphorylation occurs after DNA damage and is consistent with a prior observation[36]. The structures and biochemical analyses presented here demon-strate that pS429 serves to enhance the activity of the homodimer.

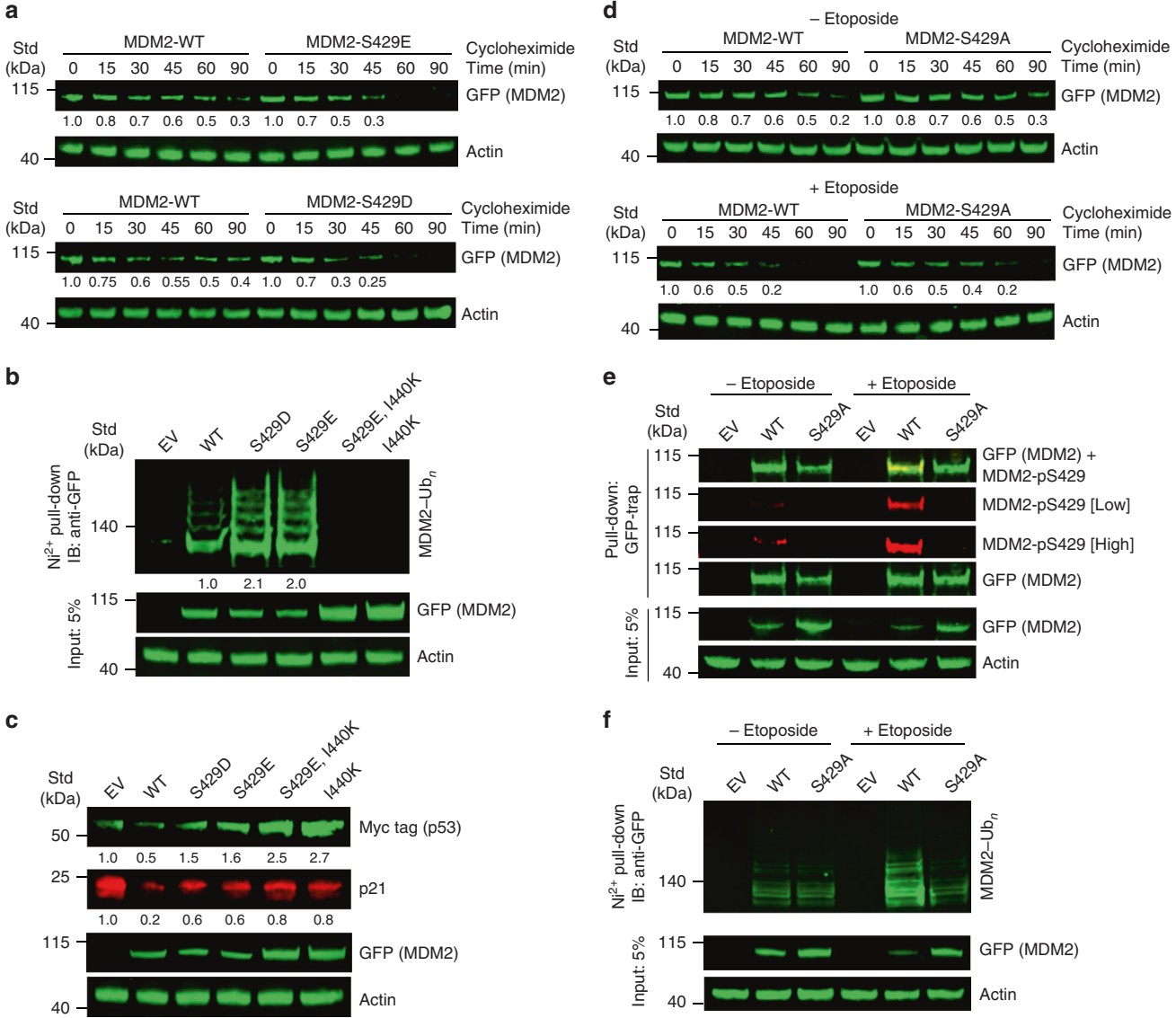

**Fig. 6 DNA damage-mediated S429 phosphorylation effects in cells. a** Immunoblots showing the stability of MDM2 variants from lysates of U2OS^mod cells expressing GFP-MDM2 variants treated with cycloheximide for indicated times. The immunoblots were analyzed by anti-GFP or anti-actin antibodies as indicated. **b** Immunoblots of MDM2 ubiquitination from lysates of U2OS^mod cells transfected with plasmids expressing GFP-MDM2 variants or empty vector (EV) along with His-Ub and treated with MG132. The cell lysates and Ni-NTA pull-down products were analyzed by immunoblotting with anti-GFP or anti-actin antibodies as indicated. **c** Immunoblots showing the effects of MDM2 variants on p53 and p21. Unmodified U2OS cells were transfected with plasmids expressing GFP-MDM2 variants or EV and Myc-tagged p53. Lysates were analyzed by immunoblotting using anti-GFP, anti-Myc tag, anti-p21, or anti-actin antibodies as indicated. **d** Immunoblots showing the stability of MDM2 variants from lysates of U2OS^mod cells expressing GFP-MDM2 variants left untreated (top panel) or treated with etoposide (bottom panel) for 6 h, followed by cycloheximide treatment for indicated times. The immunoblots were analyzed by anti-GFP or anti-actin antibodies as indicated. **e** Immunoblots showing MDM2 S429 phosphorylation in the absence and presence of etoposide treatment. U2OS^mod cells were transfected with plasmids expressing GFP-MDM2 variants or EV and treated with etoposide where indicated. The cell lysates and GFP-Trap pull-down products were analyzed by immunoblotting with anti-GFP, anti-MDM2-pS429, or anti-actin antibodies as indicated. Low = low exposure; high = high exposure. **f** Immunoblots of MDM2 ubiquitination from lysates of U2OS^mod cells transfected with plasmids expressing GFP-MDM2 variants or EV, along with His-Ub in the absence and presence of etoposide. Prior to harvesting, cells were treated with MG132. The cell lysates and Ni-NTA pull-down products were analyzed by immunoblotting with anti-GFP or anti-actin antibodies as indicated. Actin loading control blots were included for all panels. All the experiments were performed in triplicate with similar results. Raw data are provided in Supplementary Fig. 10.

We anticipate that pS429 would enhance MDM2-catalyzed ubiquitination reactions, including autoubiquitination and substrate ubiquitination in cells. However, enhanced autoubiquitination destabilizes MDM2 and presumably leads to substrate stabilization. Consistent with this notion and as observed previously[36], we showed that p53 is stabilized in the presence of MDM2-S429 phosphomimetics as compared to MDM2-WT. However, if MDM2 engages with substrates other than p53, phosphorylation

of S429 could enhance substrate ubiquitination. MDM2 has been shown to associate with Nbs1 of the Mre11-Rad50-Nbs1 complex upon DNA damage to inhibit DNA repair[59,60], but it remains unclear whether its E3 activity is associated with this process. Future studies on p53-independent functions of MDM2 after DNA damage could reveal additional roles of S429 phosphorylation. Interestingly, we observed basal S429 phosphorylation without etoposide treatment. This suggests that it might have a

role under unstressed conditions and will require further investigation.

In conclusion, our study provides structural insights into MDM2 phosphorylation near the RING domain and reveals a distinct role of S429 phosphorylation in stabilizing E2–Ub to stimulate self-degradation upon DNA damage. The findings presented here will assist in unveiling the molecular basis of MDM2 phosphoregulation, which is an important aspect in diagnostics and might serve as a target in drug design.

## Methods

**Recombinant protein preparation.** All constructs were generated by standard PCR-ligation techniques and verified by DNA-sequencing. MDM2 variants were cloned into pGEX4T1 vector containing an N-terminal GST-tag, followed by a tobacco etch virus (TEV) cleavage sequence. MDMX variants were cloned into pRSFDuet-1 vector with an N-terminal 12× His-tag, followed by a TEV cleavage site. A G443T substitution was included only in cat MDM2 variants for structure determination to improve the solubility of MDM2 after TEV cleavage. G443T has minimal effects on the folding of MDM2 RING domain or UbcH5B–Ub binding as revealed by our structures (Supplementary Fig. 8). All non-phosphorylated MDM2 variants were expressed in *Escherichia coli* BL21(DE3) GOLD. For the MDM2-MDMX heterodimer, BL21(DE3) GOLD cells were co-transformed with MDM2 and MDMX constructs. Cells were grown at 37 °C in Luria-Bertani medium to an $OD_{600}$ of 0.6–0.8 and induced with 0.2 mM isopropyl β-D-1-thiogalactopyranoside (IPTG) at 18–20 °C overnight. Cells were lysed in 50 mM Tris-HCl, pH 7.6, 0.4 M NaCl, 1 mM dithiothreitol (DTT), and 2.5 mM phenylmethylsulfonyl fluoride (PMSF). For SPR analyses and in vitro activity assays, GST-MDM2 variants were purified by Glutathione Sepharose (GE Healthcare) affinity chromatography, followed by size-exclusion chromatography to isolate dimeric MDM2 (Supplementary Fig. 9) and GST-MDM2-His-MDMX variants were purified by Ni-NTA resin, followed by Glutathione Sepharose affinity chromatography. For protein crystallization and activity assays of cleaved MDM2 (Supplementary Fig. 9), GST-MDM2 variants were subjected to TEV cleavage and further purified by size-exclusion chromatography using a HiLoad Superdex 75 26/60 (GE Healthcare) gel filtration column. To generate MDM2-pS429, EcAR7 cells were co-transformed with an MDM2 construct containing the codon for S429 replaced with a TAG codon (Addgene 52055) and a pKD-SepRS-EFSep-5xtRNASep plasmid (Addgene 52054)[46]. Cells were grown at 30 °C in Terrific Broth supplemented with 2 mM phosphoserine at an $OD_{600}$ of 2.0 and induced with 0.5 mM IPTG at 25 °C for 20 h. Cells were lysed in 50 mM Tris-HCl, pH 7.6, 0.4 M NaCl, 0.05 M NaF, 1 mM $Na_3VO_4$, 1 mM DTT, and 2.5 mM PMSF. GST-MDM2-pS429 variants were purified as the native MDM2 described above. All MDM2 and MDM2-MDMX variants were buffer exchanged into 50 mM Tris-HCl, pH 7.6, 0.4 M NaCl, and 1 mM DTT. His-tagged human UBA1 and untagged *Arabidopsis thaliana* Uba1[45] were charged with GST-Ub with 5 mM $MgCl_2$ and 5 mM ATP for 2 h at 4 °C and purified by glutathione-affinity chromatography eluted with 25 mM DTT, followed by anion exchange chromatography. Untagged UbcH5B variants were expressed from pRSF_1b and purified by SP sepharose chromatography, followed by gel filtration chromatography[45]. His-tagged Ub was purified by Ni-NTA resin[45]. To generate stably conjugated UbcH5B S22R, C85K–Ub (refers to as UbcH5B–Ub)[45], His-tagged Ub, untagged UbcH5B S22R C85K, and untagged *A. thaliana* Uba1 were mixed together in 50 mM Tris-HCl, pH 9.0, 0.2 M NaCl, 10 mM $MgCl_2$, and 10 mM ATP at 30 °C for 1 day and subsequently purified by Ni-NTA affinity chromatography, followed by treatment with TEV protease to remove the His-tag and further purification by cation exchange and size-exclusion chromatography using a HiLoad Superdex 75 26/60 gel filtration column. Fluorescently labeled Ub was expressed in pGEX4T1 vector as a GST-TEV-GGSC fusion construct. After TEV cleavage, the protein was subjected to size-exclusion chromatography using a HiLoad Superdex 75 26/60 gel filtration column equilibrated in phosphate-buffered saline (PBS) buffer containing 1 mM TCEP (tris(2-carboxyethyl)phosphine), incubated with IRDye® 800CW Maleimide (LI-COR) for 3 h at 20 °C (five-fold excess of Ub) and desalted twice using Zeba™ Spin Desalting Columns (Thermo Fisher). Protein concentrations were determined by absorbance at 280 nm for Ub variants and Bio-Rad protein assay with bovine serum albumin (BSA) as a standard for other proteins.

**SPR binding analyses.** SPR experiments were performed at 25 °C on a Biacore T200 instrument using a CM-5 chip (GE Healthcare) with coupled anti-GST antibody[44]. GST-tagged MDM2 variants were coupled on the chip and a serial dilution of UbcH5B–Ub in running buffer containing 25 mM Tris-HCl, pH 7.6, 150 mM NaCl, 0.1 mg ml⁻¹ BSA, 1 mM DTT, and 0.005% (v/v) Tween-20 was used as analyte. Two technical replicates were performed and data were analyzed using Biacore T200 BIAevaluation (GE Healthcare) and Scrubber2 (BioLogic Software) after background subtraction (GST alone).

**Autoubiquitination assays.** Autoubiquitination assays were performed at 23 °C. UbcH5B (5 μM) was pre-charged with fluorescently-labeled Ub (50–80 μM) in the

presence of UBA1 (0.2 μM) in 50 mM Tris-HCl, pH 7.6, 50 mM NaCl, 5 mM $MgCl_2$, and 5 mM ATP for 15–30 min. The reaction was then initiated by the addition of GST-MDM2 (0.8 μM) or cleaved MDM2 variants (3 μM) in PBS buffer, stopped after 0 and 90 s or indicated time points with 4× LDS loading dye containing 400 mM DTT and resolved by sodium dodecyl sulfate-polyacrylamide gel, followed by visualization with an Odyssey CLx Imaging System (LI-COR Biosciences) and staining with InstantBlue. Three independent reactions were performed. Concentrations in parenthesis indicate the final concentration in the assay. The ubiquitinated MDM2 species were quantified using the LI-COR Image Studio Litesoftware (version 5.2.5). Background subtraction was performed using the zero time point from each reaction and subsequently normalized to the amount of E3 loaded based on quantification of the InstantBlue-stained gel. For Figs. 1b, d and 4c, j, l, n, three independent reactions were loaded on the same gel for quantification. For Figs. 1f, 3g, and 5g, i, k, three independent reactions were loaded on different gels, but processed in parallel, and data were normalized across replicates from different gels based on the zero time point from each replicate. Ubiquitination activity relative to WT and standard deviation are presented.

**Crystallization.** All crystals were screened by sitting drop vapor diffusion technique and where required optimized by hanging drop vapor diffusion method by mixing protein and reservoir solution at 1:1 ratio at 19 °C.

Cat MDM2-422–C-pS429-UbcH5B–Ub complex: Cat MDM2-422–C-pS429 (5.1 mg ml⁻¹), and UbcH5B–Ub (14.1 mg ml⁻¹) were mixed in a 1:1 molar ratio. Crystals were obtained in a condition containing 0.1 M Tris-HCl, pH 8.0, 15% (w/v) PEG 2000 MME and 0.1 M KCl, and flash-frozen in 0.1 M HEPES, pH 8.0, 27% (w/v) PEG3350, 0.2 M NaCl, and 25% (v/v) ethylene glycol.

Cat MDM2-422–C-S429E-UbcH5B–Ub complex: Cat MDM2-422–C-S429E (14.9 mg ml⁻¹) and UbcH5B–Ub (14.5 mg ml⁻¹) were mixed in a 1:1 molar ratio. Crystals were obtained in a condition containing 0.1 M Tris-HCl, pH 8.5, 20% (v/v) PEG Smear High, and flash-frozen in 0.1 M Tris-HCl, pH 8.5, 20% (w/v) PEG8000, 0.2 M $NH_4NO_3$, and 25% (v/v) ethylene glycol.

Human MDM2-419–C-UbcH5B–Ub complex: Human MDM2-419–C (7.3 mg ml⁻¹) and UbcH5B–Ub (14.5 mg ml⁻¹) were mixed in a 1:1 molar ratio. The crystals were initially obtained in a condition containing 0.1 M BICINE, pH 9.3, and 22% (v/v) PEG Smear Broad and subsequently used as micro-seeds for optimization. The final crystal was grown in 0.1 M SPG, pH 7.0, 10% (w/v) PEG Smear Broad, 0.15 M $NH_4NO_3$, and flash-frozen in 0.1 M SPG, pH 7.0, 13% (w/v) PEG Smear Broad, 0.12 M $NH_4NO_3$, and 25% (v/v) glycerol.

Cat MDM2-422–C-S429E: Crystals of Cat MDM2-422–C-S429E (14.9 mg ml⁻¹) were obtained in a condition containing 0.1 M MMT, pH 9.0, and 25% (w/v) PEG1500 and flash-frozen in 0.1 M Tris-HCl, pH 8.5, 20% (w/v) PEG8000, 0.2 M $NH_4NO_3$, and 25% (v/v) ethylene glycol.

**Structure determination.** Data collection was carried out at beamlines I03, I04, and I04-1 at Diamond Light Source (DLS). Datasets were processed by automated **X-ray Detector Software** (XDS)[61] pipeline and reduced with fast_dp package for cat MDM2-pS429-UbcH5B–Ub complex or XIA2 package[62] for all other datasets. Initial phasing was accomplished by molecular replacement with PHASER[63] using the coordinates for UbcH5B, Ub, and MDM2 from PDB ID: 5MNJ. The atomic models were built with COOT[64] and refined in REFMAC5[65] and PHENIX[66]. The final model for cat MDM2-422–C-pS429-UbcH5B–Ub complex was refined to 1.83 Å and contained two copies of MDM2 (chains A and D residues 428–491), two copies of UbcH5B (chains B and E residues 2–147), and two copies of Ub (chains C and F residues 1–76). The final model for human MDM2-419–C-UbcH5B–Ub complex was refined to 1.41 Å and contained two copies of MDM2 (chains A and D residues 429–491), two copies of UbcH5B (chains B and E residues 2–147), and two copies of Ub (chains C and F residues 1–76). The final model for cat MDM2-422–C-S429E-UbcH5B–Ub complex was refined to 2.18 Å and contained four copies of MDM2 (chain A residues 423–491, chains D and G residues 428–491, and chain J residues 429–491), four copies of UbcH5B (chains B, E, H, and K residues 2–147), and four copies of Ub (chains C, F, I, and L residues 1–76). The final model for cat MDM2-422–C-S429E was refined to 1.21 Å and contained four copies of MDM2 (chain A residues 429–C, chains B and D residues 424–491, and chain C residue 427–C). Where the electron density was insufficient for modeling side chains, alanine stubs were built instead. Statistics for the final refinement are shown in Table 2. Polder density maps[67] were calculated in PHENIX to confirm the region surrounding pS429 and S429E. All figures were created with PYMOL (Schrödinger).

**Mammalian plasmids, chemicals, and antibodies.** Full-length human MDM2-WT was cloned into pGZ21dxZ-GFP, and using site-directed mutagenesis, the following mutants were generated: MDM2-S429A, MDM2-S429D, MDM2-S429E, MDM2-I440K, and MDM2-S429E-I440K. The following chemicals were used: cycloheximide (Sigma-Aldrich, cat. no. C1988), MG132 (Sigma-Aldrich, cat. no. 474787) and etoposide (Sigma-Aldrich, cat. no. E1383). Cycloheximide was dissolved in water and used at a final concentration of 50 μg ml⁻¹, while etoposide and MG132 were dissolved in dimethyl sulfoxide and used at final concentrations of 50 μM each. The cells were treated with etoposide for 6 h and with MG132 for 4 h prior to harvesting. The primary antibodies used in this study include mouse

anti-GFP (Santa Cruz Biotechnology, cat. no. sc-81045, 1:1000 for Western blot), customized rabbit anti-MDM2-pS429 (Eurogentec, 1:1000 for Western blot), mouse anti-Myc tag (Cell Signaling Technology, cat. no. 2276, 1:1000 for Western blot), rabbit anti-p21 (Cell Signaling Technology, cat. no. 2947, 1:1000 for Western blot), and goat anti-actin (Santa Cruz Biotechnology, cat. no. sc-1616, 1:1000 for Western blot). Anti-MDM2-pS429 antibody was generated by Eurogentec; Ac-KEESVESpSLPLN-CONH$_2$ peptide was used for rabbit immunization. Subsequently, anti-MDM2-pS429 antibody was purified by double affinity columns with enrichment on modified peptide, followed by depletion against unmodified peptide. The secondary antibodies used were goat anti-mouse IRDye 800CW (LI-COR Biosciences, cat. no. 925-32210, 1:15000 for Western blot), goat anti-rabbit IRDye 680LT (LI-COR Biosciences, cat. no. 925-68021, 1:20,000 for Western blot), and donkey anti-goat IRDye 800CW (LI-COR Biosciences, cat. no. 925-32214, 1:15,000 for Western blot). GFP-Trap was purchased from Chromotek.

**Mammalian cell maintenance and transfections**. Human osteosarcoma U2OS cells (ATCC) with *MDM2*-knockout and doxycycline-inducible p53 shRNA (U2OS$^{mod}$)[44] and unmodified U2OS cells were cultured in Dulbecco's modified Eagle's medium, supplemented with 10% fetal bovine serum, 0.1 mg ml$^{-1}$ streptomycin, 100 U ml$^{-1}$ penicillin, 20 mM L-glutamine, and 6 mg l$^{-1}$ gentamicin reagent solution (Invitrogen, USA). For U2OS$^{mod}$ cells, the medium was further supplemented with 50 µg ml$^{-1}$ doxycycline to keep p53 knocked down. The cells were cultured in monolayer at 37 °C in 5% CO$_2$ and transfected using jetPRIME transfection reagent (Polyplus transfection) following the manufacturer's protocol. For cycloheximide chase experiments (Fig. 6a, d), U2OS$^{mod}$ cells were transfected with 2.5 µg MDM2 variants. Cell-based ubiquitination assays (Fig. 6b, f) were performed by transfecting U2OS$^{mod}$ cells with 5 µg MDM2 variant or EV as indicated in the figures, along with 1 µg 12× His-Ub. In Fig. 6c, U2OS cells were co-transfected with 1 µg Myc-p53 and 5 µg of the indicated MDM2 variant or EV, while in Fig. 6e, U2OS$^{mod}$ cells were transfected with 5 µg MDM2 variant or EV as indicated. Cells were harvested 36 h post transfection.

**Cycloheximide chase assay**. U2OS$^{mod}$ cells were transfected with constructs as shown in Fig. 6a, d and maintained for 36 h. Cells were then treated with fresh medium containing 50 µg ml$^{-1}$ cycloheximide and harvested at time points as indicated in Fig. 6a, d. Whole-cell lysates were prepared and the stability of MDM2 variants were checked through immunoblotting.

**Preparation of lysate for analyses**. Whole-cell lysates were prepared in 50 mM Tris-HCl, pH 7.2, 150 mM NaCl, 10% (v/v) glycerol, 1% (v/v) IGEPAL CA-630, and freshly supplemented with protease inhibitor cocktail, 2.5 mM PMSF, 0.5 mM DTT, and 1 mM EDTA. For immunoprecipitation (IP), 0.5 mg of freshly prepared lysates were incubated with 35 µL (50% slurry) GFP-Trap for 1 h at 4 °C on a rotatory shaker. The beads were then washed twice with IP lysis buffer and once with IP wash buffer (IP lysis buffer modified to contain 200 mM NaCl and 1 mM DTT). The beads were eluted by incubating at 95 °C for 10 min with 40 µL 2× protein loading dye. For Western blotting of whole-cell lysates, 50 µg protein was loaded per lane. Protein samples were separated by SDS-PAGE in NuPAGE™ 4–12% Bis-Tris gels (Thermo Fisher Scientific) using MES-SDS running buffer and then transferred onto nitrocellulose membranes from Trans-Blot® Turbo™ Transfer Packs (Bio-Rad) using a Trans-Blot® Turbo™ Transfer System (Bio-Rad). The blots were incubated with the indicated primary antibodies at 4 °C overnight and visualized after incubation with secondary antibodies using an Odyssey CLx Imaging System. Immunoblots for Fig. 6a–d were quantified using the LI-COR Image Studio Lite software. Band intensities from each lane were quantified and then normalized using actin band intensities. For Fig. 6a, d, assigned values for each band intensity are relative to the zero time point of the corresponding condition, and for Fig. 6c, values for each band are relative to EV.

**Cell-based ubiquitination assay**. Cells were transfected with plasmids expressing His-Ub and full-length GFP-MDM2 variants as indicated. Thirty-six hours after transfection, cells were harvested following treatment with 50 µM MG132 for 4 h. Whole-cell lysates were prepared in IP lysis buffer (without DTT) with three 20 s cycle of sonication at 20% amplitude. Protein concentrations were measured using a Bradford assay and 0.5 mg total protein for each set was normalized for volume and then mixed with Ubiquitination buffer A (UBA) buffer (6 M GuHCl, 0.3 M NaCl, 50 mM phosphate pH 8.0, 100 µg ml$^{-1}$ NEM) to a final volume of 1 mL. Dynabeads His-tag matrices (Invitrogen) were added to the lysates and incubated on a rotatory shaker at 4 °C overnight. The next day, the beads were sequentially washed with UBA, Ubiquitination buffer B (UBB), Ubiquitination buffer C (UBC), and PBS (UBB = UBA and UBC 1:1; UBC = 0.3 M NaCl, 50 mM phosphate, pH 8.0, 100 µg ml$^{-1}$ NEM) and eluted in 2× protein loading dye by incubating at 95 °C for 10 min. The eluted samples were resolved by SDS-PAGE and immunoblotted with the antibodies as indicated in Fig. 6. The ubiquitinated MDM2 variant species were quantified using the LI-COR Image Studio Lite software and values assigned for each band are relative to WT.

**MS analysis**. GST-MDM2-419–C-pS429 was reduced with 10 mM DTT and alkylated with 55 mM iodoacetamide and finally digested with trypsin (Promega). Digested peptides were desalted using StageTip[68]. Tryptic peptides were separated by nanoscale C18 reverse-phase liquid chromatography using an EASY-nLC 1200 (Thermo Fisher Scientific) coupled online to an Orbitrap Q-Exactive HF mass spectrometer (Thermo Fisher Scientific) via nanoelectrospray ion source (Thermo Fisher scientific). Peptides were separated on a 50 cm fused silica emitter (New Objective) packed in house with reverse-phase ReprosilPur Basic 1.9 µm (Dr. Maisch GmbH). For the full scan, a resolution of 60,000 at 250 Th was used. The top ten most intense ions in the full MS were isolated for fragmentation with a target of 50,000 ions at a resolution of 15,000 at 250 Th. MS data were acquired and processed using the XCalibur software (Thermo Fisher Scientific). The MS Raw files were processed with the MaxQuant software[69] (version 1.5.5.1) and searched with Andromeda search engine[70], querying UniProt and a fasta database containing the MDM2 modified sequences used in the experiment. Database was searched requiring specificity for trypsin cleavage and allowing maximum two missed cleavages. The iodoacetamide derivative of cysteine was specified as a fixed modification. Protein N-terminal acetylation, oxidation of methionine, and phosphorylation of serine, threonine, and tyrosine were all specified as variable modifications. The peptide, protein, and site false discovery rate was set in MaxQuant to 1%. Scaffold (version 4.8.4, Proteome software Inc., Portland, OR) was used to validate MS/MS-based peptide and protein identifications and to visualize the MS/MS spectra.

**Reporting summary**. Further information on research design is available in the Nature Research Reporting Summary linked to this article.

## Data availability
Atomic coordinates and structure factors are deposited in the Protein Data Bank with accession codes of 6SQO (human MDM2-419–C–UbcH5B–Ub complex), 6SQS (cat MDM2-422–C-pS429-UbcH5B–Ub complex), 6SQR (cat MDM2-422–C-S429E-UbcH5B–Ub complex), and 6SQP (cat MDM2-422–C-S429E). Raw data for Fig. 1a, c, e are provided in Supplementary Fig. 3, raw data for Figs. 3f, 4b, i, k, m, and 5f, h, j are provided in Supplementary Fig. 5 and raw data for Fig. 6 are provided in Supplementary Fig. 10. All data are available from the authors upon reasonable request.

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

## Acknowledgements

We thank Core Services and Advanced Technologies at the Cancer Research UK Beatson Institute (C596/A17196), with particular thanks to Proteomics and Molecular Technologies services, Lori Buetow for comments, Catherine Winchester for critical reading of the manuscript, and DLS for access to stations I03, I04, and I04-1 (BAG allocation mx16258). This work was supported by Beatson Institute core (A17196), Cancer Research UK (A23278), and European Research Council (ERC) under the European Union's Horizon 2020 research and innovation program (grant agreement no. 647849) awarded to D.T.H., by the Intramural Research Program of the National Institutes of Health, National Cancer Institute, Center for Cancer Research (V.A.H. and A.M.W.), and by NIH Grant GM065334 to D.F. L.J.A. was funded by an EPSRC studentship (EP/247 M506539/1 and EP/N509668/1). K.H.V. was funded by Cancer Research UK Grants C596/A26855 and supported by the Francis Crick Institute, which receives its core

funding from Cancer Research UK (FC0010557), the UK Medical Research Council (FC0010557), and the Wellcome Trust (FC0010557).

## Author contributions

H.M.M. performed in vitro biochemical assays, protein crystallization, and X-ray data collection. H.M.M., L.J.A., A.G.J., and D.T.H. designed and generated MDM2-pS429. H.M.M. and D.T.H. performed protein purification and structure determination. S.F.A., K.N., A.K.H., and K.H.V. performed mammalian cell assays and analyzed the data. V.A.H., D.F., and A.M.W. performed initial biochemical analyses. H.M.M. and G.J.S. performed and analyzed SPR experiments. H.M.M., S.F.A., and D.T.H. wrote the manuscript. All authors read and approved the final manuscript.

## Competing interests

K.H.V. is on the Board of Directors and shareholder of Bristol Myers Squibb, a shareholder of GRAIL Inc. and on the Science Advisory Board (with stock options) of PMV Pharma, RAZE Therapeutics, and Volastra Therapeutics. She is also on the SAB of Ludwig Cancer. K.H.V. is a co-founder and consultant of Faeth Therapeutics, funded by Khosla Ventures. She has been in receipt of research funding from Astex Pharmaceuticals and AstraZeneca and contributed to CRUK Cancer Research Technology filing of Patent Application WO/2017/144877.
