## [Peer Review File · Nature Communications]

Reviewers' comments:

Reviewer #1 (Remarks to the Author):

This manuscript delineates the structural basis of how the phosphorylation of a particular site (Ser429) enhances the ligase activity of the MDM2 homodimer (but not the MDM2-MDMX heterodimer) to promote auto-ubiquitination and stabilize P53. The work is important, novel, and highly interesting and provides a wealth of structural and functional data with interesting implications for our understanding of MDM2-P53 interplay. However, in my eyes, a number of points require improvement before publication:

- (1) All in vitro activity assays (1A, 1C, 1E, 3F, 4B, I, K, M, 5G, I K) are lacking important controls, demonstrating that comparable amounts of ligase were used in the individual reactions (w/o ATP). It is necessary to show the input ligase +/- ATP (e.g. as Coomassie stain) and normalize the amount of product to the input in the quantifications.
- (2) It would be important to show - at least in Figure 1A - an uncropped gel (both as fluorescence image and Coomassie) in order to demonstrate that the bands above 35 kDa are not impacted by other reactions (i.e. free chain formation and E2-ubiquitination).
- (3) It is unclear how the quantification in Figure 1F was done, given that the corresponding data in Figure 1E are on separate gels. How were the data normalized?
- (4) When comparing the WT lanes in Figure 5G and 5I/K it appears that the image in Figure 5G was exposed much less. The authors should show 5G at a similar exposure as I/K. This is important, since the mutant S429E_Ala_ins in 5G shows longer chains than WT_Ala_ins, however, this difference is not visible in the quantification.
- (5) SPR analyses: It is not fully clear to me how the SPR analysis was performed in terms of the different stoichiometries of the interactions of the MDM2 homodimer and the MDM2-MDMX heterodimer, respectively, with the E2-conjugate. Was the concentration of binding sites (MDM2) taken into account for the K_d-values reported in Table 1? If yes, please clarify in the text and figure legends/titles that the reported K_d refers to one MDM2 subunit in the context of either a homo or a heterodimer.
- (6) The assays in Figures 6A, C, and D require quantification, since quantitative statements are being made, which are not easy to extract from the images by eye (considering the effects of protein levels, loading etc.).
- (7) The authors comment on trace amounts of MDM2-pS429 in Figure 6E, however, I can not spot them in the figure.

Reviewer #2 (Remarks to the Author):

In this manuscript, Magnussen et al. describe a series of crystal structures of homodimeric MDM2 RING E3 ubiquitin ligase which is a crucial regulator of p53 levels in the cell. The central theme of the paper relates to the discovery that phosphorylation of Ser429 of MDM2 leads to an increase in the autoubiquitination activity of MDM2 homodimer by stabilizing the 'closed' E2~Ub conformation that is primed for Ub discharge. Given that ATM has been shown to phosphorylate Ser429 of MDM2, the authors propose that this phosphorylation-induced elevation of MDM2 autoubiquitination activity is a mechanism by which p53 levels are stabilized in the cell in response to DNA damage. Comparison to previously reported MDM2/MDMX heterodimer structures reveals novel structural features in the N-terminal region of MDM2 molecules in the homodimer that position pSer429 in proximity of Ub and

thereby facilitating contacts that account for the increase in activity. pSer429 is distant from Ub in MDM2/MDMX heterodimer structures which is proposed to account for why phosphorylation of MDM2 Ser429 fails to alter the activity of the heterodimer. The authors performed a series of biochemical and cell-based experiments that largely support conclusions made based on their structures and all of the data presented in the manuscript appear to be of high quality. Overall, this study will be of interest to a wide range of readers who are interested in the DNA damage response, cell cycle regulation, ubiquitin signaling, and tumorigenesis, and as such, this reviewer recommends publication in Nature Communications provided that the following minor concerns are addressed:

- * Can the authors explain why feline MDM2 crystalized with pSer429 (and phosphomimetics) but human failed to? This is somewhat surprising given the very high sequence conservation throughout the protein except for the N-terminus (around Ser429). Are there crystal contacts in this region that are mediated by residues unique to feline MDM2?
- * Since the reported phosphoregulatory mechanism only pertains to the MDM2 homodimer, it would be helpful if the authors highlight what is known about the importance of MDM2 homodimer activity as it compares to MDM2/MDMX2 heterodimer activity in the cell. This would help better frame the biological relevance of the author's findings.
- * How did the authors decide to use the G443T mutant version of MDM2? Does the structure account for the observed increase in solubility? Does the G443T mutant exhibit the same levels of activity as WT (I presume WT was used in the biochemical assays)? This information should be included in the manuscript.
- * The authors should present composite omit maps for the regions shown in Figure 4e,g.
- * Along these lines, the authors state that their pSer429 sample is not homogenous. Were efforts made to determine the ratio of phosphorylated to unphosphorylated MDM2? Was the occupancy of the phosphate of pSer429 refined during model building, and if so, what is the occupancy? If not, it might be useful to refine occupancies, especially at the reported resolution, as this information would be useful to report.
- * The layout of the panels in figure 5 is confusing.
- * It would be helpful if the data presented in Figure 6 was quantified.
- * Typo on page 14, line 277 (asymmetrical). The manuscript should be carefully reviewed for typos and the used of commas.

Reviewer #3 (Remarks to the Author):

Mdm2 is a Ubiquitin E3 ligase that regulates the stability of the tumor suppressor protein p53. The manuscript "Structural basis for DNA damage-induced phosphoregulation of MDM2 RING domain" by

Magnussen and colleagues builds on a report that ATM phosphorylates Mdm2 at six positions to regulate p53 stability. Magnussen and colleagues use biochemistry and crystallographic data to demonstrate that the phosphorylation of Mdm2 at one of such position, serine 429, increases the binding of the Mdm2 homodimer (but not the Mdm2-MdmX heterodimer) to UbcH5b~Ub, resulting in enhanced Ubiquitin E3 ligase activity. They proceed in showing that this occurs through the formation of a new interaction between the phosphate moiety at position S429 and residue K33 of Ubiquitin in the closed conformation. They use structural data to assess why this effect is specific to Mdm2 homodimers and show that the phosphorylation-mediated increase in Ubiquitin E3 ligase activity translates in vivo with a phosphomimetic version of Mdm2 displaying a shorter half-life and stabilizing p53. They finally show that, in vivo, the phosphorylation of Mdm2 at position 429 is enhanced upon DNA damage, leading to quicker Mdm2 degradation.

Overall, the authors presented solid biochemistry and compelling structural evidence that provide a molecular understanding of the role of Serine 429 phosphorylation on Mdm2 activity and stability. They use different strategies (point mutants, insertion mutants and charge swapping mutants) to establish (i) that the phosphorylation of Serine 429 of Mdm2 indeed contacts the donor Ubiquitin and (ii) why this mechanism can only happen in the context of the Mdm2 homodimer. The manuscript is well written, the methods are well detailed, the figures are clear and the authors pay great care not to oversell their claims. I foresee that this article will be of significant interest for different communities working on ubiquitin, p53 and DNA damage and, therefore, deserves publication in Nature Communications. However, I have two concerns about the manuscript.

Major points:

1. The article does not address the question of the oligomerization status of Mdm2 upon S429 phosphorylation. Although the structure of the phosphorylated Mdm2 dimer does not point into that direction, I am a bit worried about the effect of phosphorylation on the oligomerization status of Mdm2 and the use of GST-tagged Mdm2 proteins for Ubiquitination assays. As referenced by the authors (lines 83-85), there is evidence from Cheng, EMBO J, 2009 and Cheng, Mol Cell Biol, 2011 that Mdm2 phosphorylation, including at the 429 position, weakens Mdm2 dimerization/oligomerization. In Supplementary Figure 2, an oligomerization ladder even appears to be present with GST-Mdm2 but is absent with GST-Mdm2 p429 (I am comparing lane 1 and 2 in the InstaBlue stained gel). Use of GST-fusions in the present manuscript may result in shifted oligomerization equilibrium (due to GST dimerization) that could distort the results. Do the authors have evidence that the phosphorylation of Mdm2 at position 429 affect its oligomerization status? Alternatively, can the authors provide evidence that the main conclusions of their Ubiquitination assays hold true when using untagged/cleaved proteins?

2. Can the authors discuss the role of symmetry-related interactions on the stabilization of 3-10 vs alpha helices for Mdm2 and MdmX? Indeed, in their previous structure of Mdm2-MdmX-UbcH5B~Ub (pdb 5MNJ), there is an interaction between a symmetry-related UbcH5B molecule and residues 430-436 of Mdm2 that fold into an alpha helix. As interactions with symmetry-related molecules could contribute to the stabilization of one particular type of secondary structure element at the N-terminus of Mdm2 or MdmX, I believe it would be important for the authors to take those interactions into account.

Minor points:

1. Could the authors indicate the distances for the hydrogen bonds depicted in Figure 4f and 4h.
2. It would be interesting to directly compare the results obtained with the I440K S429E mutant to the

ones obtain with the I440K mutant in Figure 6b,c to show that there are no other effects of S429E in the context of a “dead ligase”.

3. In Figure 6d, it appears Mdm2 S429A is less stable than Mdm2 WT in the absence of Etoposide (no band at 90 min for S429A whereas a band is present for WT). Could the authors comment on this?

We would like to thank the reviewers for their comments. All responses are in blue below.

Reviewer #1 (Remarks to the Author):

This manuscript delineates the structural basis of how the phosphorylation of a particular site (Ser429) enhances the ligase activity of the MDM2 homodimer (but not the MDM2-MDMX heterodimer) to promote auto-ubiquitination and stabilize P53. The work is important, novel, and highly interesting and provides a wealth of structural and functional data with interesting implications for our understanding of MDM2-P53 interplay. However, in my eyes, a number of points require improvement before publication:

(1) All in vitro activity assays (1A, 1C, 1E, 3F, 4B, I, K, M, 5G, I K) are lacking important controls, demonstrating that comparable amounts of ligase were used in the individual reactions (w/o ATP). It is necessary to show the input ligase +/- ATP (e.g. as Coomassie stain) and normalize the amount of product to the input in the quantifications.

All E3s used in the in vitro activity assays were purified. The protein concentrations were determined by BioRAD assay and then loaded on the SDS-PAGE where the concentrations of mutant E3s were normalized relative to WT E3 band. Subsequently 100 nM E3s were used in the assay. At this concentration it was not visible on InstantBlue stain. We have now repeated these experiments at higher E3 concentration such that comparable amounts of E3 can be visualized by InstantBlue staining. Zero reaction time point is now performed to show similar E3 loading. The amount of E3s were estimated and normalized in the quantification. The uncropped and InstantBlue gels are now shown in Supplementary Figures 3 and 5. An example of +/- ATP and DTT SDS-PAGE is now included in Supplementary Figure 3d to show that the reaction is dependent on ATP and UbcH5B~Ub remained charged throughout the reaction. Thus the observed increase in activity is due to the S429E substitution.

(2) It would be important to show - at least in Figure 1A – an uncropped gel (both as fluorescence image and Coomassie) in order to demonstrate that the bands above 35 kDa are not impacted by other reactions (i.e. free chain formation and E2-ubiquitination).

We have now provided all the uncropped gels in Supplementary Figures 3 and 5. The bands above 35 kDa are ubiquitinated GST-MDM2 variants. This is further supported by the disappearance of GST-MDM2 variant band and the increase in the higher molecular weight bands in the InstantBlue stained gels.

(3) It is unclear how the quantification in Figure 1F was done, given that the corresponding data in Figure 1E are on separate gels. How were the data normalized?

The experiment in the original Figure 1E was done on the same gel, but the lane order was not ideal. Therefore we presented them separately for figure presentation and apologize for the confusion. We have now repeated this experiment with higher E3 concentration and replaced the figure (see above and Figure 1E).

For quantification of all gels, three independent reactions were loaded on the same gel where possible. The ubiquitinated MDM2 species were quantified using the LI-COR Image Studio Lite software. Background subtraction was performed using the zero time point from each reaction and subsequently normalized to the amount of E3 loaded based on quantification of the InstantBlue stained gel. In cases where three independent reactions were loaded on separate gels (Figures 3f, 5f, 5h and 5j), the data were normalized across replicates from different gels based on the zero time point from each replicate. This is now included in the method section.

(4) When comparing the WT lanes in Figure 5G and 5I/K it appears that the image in Figure 5G was exposed much less. The authors should show 5G at a similar exposure as I/K. This is important, since the mutant S429E_Ala_ins in 5G shows longer chains than WT_Ala_ins, however, this difference is not visible in the quantification.

We agree with the reviewer that in the original Figure 5G, S429E_Ala_ins appears to show slightly long chains than WT_Ala_ins. However, when we quantified, the difference was not visible. We have now repeated these Figures (now Figure 5f, 5h and 5j) with controls as suggested above and presented with similar exposure. We do not observe difference between WT_Ala_ins and S429E_Ala_ins consistent with our data that Ala insertion reduced the effect of S429E-mediated activity enhancement

(5) SPR analyses: It is not fully clear to me how the SPR analysis was performed in terms of the different stoichiometries of the interactions of the MDM2 homodimer and the MDM2-MDMX heterodimer, respectively, with the E2-conjugate. Was the concentration of binding sites (MDM2) taken into account for the K_d-values reported in Table 1? If yes, please clarify in the text and figure legends/titles that the reported K_d refers to one MDM2 subunit in the context of either a homo or a heterodimer.

In the SPR experiment, ligands (MDM2 or MDM2-MDMX) are coupled onto a CM5 chip and then analyte (UbcH5B-Ub) was continuously added to reach a steady state and removed from the chip by sample flow (see Supplementary Figure 1). Seven different analyte concentrations were applied to the chip. K_d was then determined based on analyte concentration and the corresponding response unit (or binding unit). The analysis does not take MDM2 concentration into account. While we can estimate the theoretical maximal response unit based on the known ligand concentration that we coupled to the chip, SPR does not provide reliable stoichiometry measurement. This is likely due to immobilization of protein to a chip that renders some protein molecule inaccessible to the analyte.

(6) The assays in Figures 6A, C, and D require quantification, since quantitative statements are being made, which are not easy to extract from the images by eye (considering the effects of protein levels, loading etc.).

We have now quantified these Figures.

(7) The authors comment on trace amounts of MDM2-pS429 in Figure 6E, however, I can not spot them in the figure.

We have now provided a long exposure of this image in Figure 6E.

Reviewer #2 (Remarks to the Author):

In this manuscript, Magnussen et al. describe a series of crystal structures of homodimeric MDM2 RING E3 ubiquitin ligase which is a crucial regulator of p53 levels in the cell. The central theme of the paper relates to the discovery that phosphorylation of Ser429 of MDM2 leads to an increase in the autoubiquitination activity of MDM2 homodimer by stabilizing the ‘closed’ E2~Ub conformation that is primed for Ub discharge. Given that ATM has been shown to phosphorylate Ser429 of MDM2, the authors propose that this phosphorylation-induced elevation of MDM2 autoubiquitination activity is a mechanism by which p53 levels are stabilized in the cell in response to DNA damage. Comparison to previously reported MDM2/MDMX heterodimer structures reveals novel structural features in the N-terminal region of MDM2 molecules in the homodimer that position pSer429 in proximity of Ub and thereby facilitating contacts that account for the increase in activity. pSer429 is distant from Ub in MDM2/MDMX heterodimer structures which is proposed to account for why phosphorylation of MDM2 Ser429 fails to alter the activity of the heterodimer. The authors performed a series of biochemical and cell-based experiments that largely support conclusions made based on their structures and all of the data presented in the manuscript appear to be of high quality. Overall, this study will be of interest to a wide range of readers who are interested in the DNA damage response, cell cycle regulation, ubiquitin signaling, and tumorigenesis, and as such, this reviewer recommends publication in Nature Communications provided that the following minor concerns are addressed:

* Can the authors explain why feline MDM2 crystalized with pSer429 (and phosphomimetics) but human failed to? This is somewhat surprising given the very high sequence conservation throughout the protein except for the N-terminus (around Ser429). Are there crystal contacts in this region that are mediated by residues unique to feline MDM2?

We would like to point out that it was very challenging to generate sufficient quantity of MDM2-pS429 for crystallization screen as the translational insertion of O-phosphoserine system yields mostly the truncated protein (likely ends at 429 missing the rest of the protein). We managed to purify a small quantity of human MDM2-pS429 sufficient for screening around the crystallization condition of human MDM2-419-C-UbcH5B-Ub complex, but failed to obtain any crystals. We have screened MDM2-pS429 from other species and were fortunate to obtain crystals for feline MDM2-pS429-UbcH5B-Ub complex at a similar crystallization condition. Supposedly if one could generate sufficient quantity of human MDM2-pS429 for proper crystallization screen, it would be possible to crystallize it. We have now stated “Due to the low protein yield of the O-phosphoserine system, we were only able to screen around the crystallization condition of

human MDM2-419-C-UbcH5B-Ub complex and did not obtain crystals”.

We also want to highlight that we were surprised that cat MDM2-422-C-S429E-UbcH5B-Ub and cat MDM2-422-C-pS429-UbcH5B-Ub complexes only differ by phosphoserine and glutamate at 429 and yet they packed differently. Thus subtle differences could influence their crystallization.

For cat MDM2-422-C-S429E-UbcH5B-Ub complex, residue (423-427) at the N-terminus of MDM2 was involved in the crystal contact and this is unique to feline sequence. For cat MDM2-422-C-pS429-UbcH5B-Ub complex, H432 (L432 in human) appears to stack against an arginine side chain from a symmetry-related molecule. We have now included a statement in the results section and Supplementary Figure 7b,c to show the crystal packing contacts.

* Since the reported phosphoregulatory mechanism only pertains to the MDM2 homodimer, it would be helpful if the authors highlight what is known about the importance of MDM2 homodimer activity as it compares to MDM2/MDMX2 heterodimer activity in the cell. This would help better frame the biological relevance of the author's findings.

Despite over two decades of studies, it remains challenging to separate the functions of homodimer and heterodimer activity in cells. As S429 is phosphorylated after DNA damage, our study demonstrated how this phosphorylation could enhance homodimer activity to promote its degradation and have discussed how MDM2-MDMX heterodimer activity is regulated after DNA damage in the manuscript. MDM2 has been shown to bind Nbs1 of the MRN complex after DNA damage to inhibit DNA repair, but the role of E3 activity in this process remains unclear. We have now included a statement in the discussion and suggested “future studies on p53-independent functions of MDM2 after DNA damage could reveal additional roles of S429 phosphorylation”.

* How did the authors decide to use the G443T mutant version of MDM2? Does the structure account for the observed increase in solubility? Does the G443T mutant exhibit the same levels of activity as WT (I presume WT was used in the biochemical assays)? This information should be included in the manuscript.

MDM2 RING domain was shown to aggregate during purification in the literature and we have experienced this issue notably with our previous MDM2 construct (Nomura et al NSMB 2017). Since then we have optimized our construct length and developed a purification procedure such that we could isolate the dimeric fraction for assay and structural determination of human MDM2-419-C-UbcH5B-Ub complex (see Supplementary Figure 9). However with this protocol, it requires 100 L of LB culture with a final yield of ~1-2 mg of dimeric RING domain. This poses a great challenge for making MDM2-pS429 (as mentioned above expressed mostly as truncated protein). During purification of MDM2 RING domain from other species, we identified several species that have no aggregation issue. Through systematic amino acid substitution, we found G443T greatly improved MDM2 RING domain solubility and stability. G443 or

G443T sits at a region close to the E2 binding surface but is not involved in E2 binding. We have now presented the structural comparison of human MDM2-419-C-WT and cat MDM2-422-C-G443T (r.m.s.d. of 0.27 Å) in Supplementary Figure 8. We observed similar UbcH5B–Ub binding affinity and activity as MDM2-WT. The structure does not explain the improved solubility. We have opted to present the details on identification, characterization and purification of G443T in a separate manuscript, as this will benefit the MDM2 community particularly with structural analysis (manuscript in preparation). WT version of MDM2 was used in all assays in the manuscript. G443T was only used for cat MDM2 structural studies. This is now clarified in the manuscript in the method section.

* The authors should present composite omit maps for the regions shown in Figure 4e,g.

We have now replaced Figure 4e,g with polder omit maps.

* Along these lines, the authors state that their pSer429 sample is not homogenous. Were efforts made to determine the ratio of phosphorylated to unphosphorylated MDM2? Was the occupancy of the phosphate of pSer429 refined during model building, and if so, what is the occupancy? If not, it might be useful to refine occupancies, especially at the reported resolution, as this information would be useful to report.

We could not accurately determine the ratio of phosphorylated versus unphosphorylated MDM2 by mass spectrometry as the phosphorylated and unphosphorylated peptides have different ionization efficiencies. We have included sodium fluoride and sodium vanadate to inhibit protein phosphatase during purification and this is already stated in method section of the manuscript. We did not refine the occupancy of the phosphate of pS429. At occupancy of 1.0, we did not observe negative density in the Fo-Fc map. We concluded that the fraction of unphosphorylated MDM2 could be low in the sample. Alternatively, given that cat MDM2-422–C-S429E-UbcH5B–Ub and cat MDM2-422–C-pS429-UbcH5B–Ub complexes packed differently in the crystal lattice, we cannot exclude the possibility that only pS429-complex was crystallized. We have now included a statement “However, we cannot exclude the possibility that only the phosphorylated species of cat MDM2 crystallized in complex with UbcH5B–Ub” in the results section.

* The layout of the panels in figure 5 is confusing.

We have now combined Figure 5b and 5d into 5b left and right panel.

* It would be helpful if the data presented in Figure 6 was quantified.

We have now quantified Figure 6a-d.

* Typo on page 14, line 277 (asymmetrical). The manuscript should be carefully reviewed for typos and the used of commas.

We thank the reviewer for the comment. The manuscript has now been carefully

reviewed.

Reviewer #3 (Remarks to the Author):

Mdm2 is a Ubiquitin E3 ligase that regulates the stability of the tumor suppressor protein p53. The manuscript “Structural basis for DNA damage-induced phosphoregulation of MDM2 RING domain” by Magnussen and colleagues builds on a report that ATM phosphorylates Mdm2 at six positions to regulate p53 stability. Magnussen and colleagues use biochemistry and crystallographic data to demonstrate that the phosphorylation of Mdm2 at one of such position, serine 429, increases the binding of the Mdm2 homodimer (but not the Mdm2-MdmX heterodimer) to UbcH5b~Ub, resulting in enhanced Ubiquitin E3 ligase activity. They proceed in showing that this occurs through the formation of a new interaction between the phosphate moiety at position S429 and residue K33 of Ubiquitin in the closed conformation. They use structural data to assess why this effect is specific to Mdm2 homodimers and show that the phosphorylation-mediated increase in Ubiquitin E3 ligase activity translates in vivo with a phosphomimetic version of Mdm2 displaying a shorter half-life and stabilizing p53. They finally show that, in vivo, the phosphorylation of Mdm2 at position 429 is enhanced upon DNA damage, leading to quicker Mdm2 degradation.

Overall, the authors presented solid biochemistry and compelling structural evidence that provide a molecular understanding of the role of Serine 429 phosphorylation on Mdm2 activity and stability. They use different strategies (point mutants, insertion mutants and charge swapping mutants) to establish (i) that the phosphorylation of Serine 429 of Mdm2 indeed contacts the donor Ubiquitin and (ii) why this mechanism can only happen in the context of the Mdm2 homodimer. The manuscript is well written, the methods are well detailed, the figures are clear and the authors pay great care not to oversell their claims. I foresee that this article will be of significant interest for different communities working on ubiquitin, p53 and DNA damage and, therefore, deserves publication in Nature Communications. However, I have two concerns about the manuscript.

Major points:

1. The article does not address the question of the oligomerization status of Mdm2 upon S429 phosphorylation. Although the structure of the phosphorylated Mdm2 dimer does not point into that direction, I am a bit worried about the effect of phosphorylation on the oligomerization status of Mdm2 and the use of GST-tagged Mdm2 proteins for Ubiquitination assays. As referenced by the authors (lines 83-85), there is evidence from Cheng, EMBO J, 2009 and Cheng, Mol Cell Biol, 2011 that Mdm2 phosphorylation, including at the 429 position, weakens Mdm2 dimerization/oligomerization. In Supplementary Figure 2, an oligomerization ladder even appears to be present with GST-Mdm2 but is absent with GST-Mdm2 p429 (I am comparing lane 1 and 2 in the InstaBlue stained gel). Use of GST-fusions in the present manuscript may result in shifted oligomerization equilibrium (due to GST dimerization) that could distort the results. Do

the authors have evidence that the phosphorylation of Mdm2 at position 429 affect its oligomerization status? Alternatively, can the authors provide evidence that the main conclusions of their Ubiquitination assays hold true when using untagged/cleaved proteins?

MDM2 has been shown to oligomerize or aggregate in the literature and we have seen that during our protein purification. However, we noticed that different construct lengths and purification procedures (including buffer and protein concentration) could influence the dimeric versus the oligomeric populations. Thus one needs to be cautious in interpreting oligomerization status. With MDM2-419-C, we observed less oligomerization as compared to MDM2-428-C in our previous study (Normura et al. 2017 NSMB). In our current manuscript we purified the dimeric form of MDM2 for the assays and structural determination. GST-MDM2 RING, cleaved MDM2 RING and S429E counterparts behave as dimer on gel filtration chromatography. This is now included in Supplementary Figure 9 and we stated the purification of the dimer in the method. We did not observe evidence of S429 phosphorylation/phosphomimetic affecting its oligomerization during purification.

Supplementary Figure 2 left panel is a SDS-PAGE. Lane 2 GST-MDM2 was purified by glutathione sepharose-affinity chromatography without further gel filtration purification to illustrate this protein is not phosphorylated. The upper bands are contaminant and cannot be non-covalent oligomer since it is a SDS-PAGE. We have now stated this protein was purified after glutathione sepharose-affinity chromatography without further purification in the Figure legend for clarification.

GST was used in the assay because MDM2-RING domain lacks accessible lysine residues for efficient autoubiquitination. GST was included as the acceptor for better visualization and quantification. We have now included an assay where we used cleaved MDM2 RING and showed that S429E still enhanced the activity compared to WT (Supplementary Figure 3b,c). In the absence of GST, Ub was being utilized as the substrate to generate di-Ub, but this band overlapped with traced background maleimide-labelled Ub. For this reason we opted to use GST-MDM2 RING in our assay for better visualization.

2. Can the authors discuss the role of symmetry-related interactions on the stabilization of 3-10 vs alpha helices for Mdm2 and MdmX? Indeed, in their previous structure of Mdm2-MdmX-UbcH5B~Ub (pdb 5MNJ), there is an interaction between a symmetry-related UbcH5B molecule and residues 430-436 of Mdm2 that fold into an alpha helix. As interactions with symmetry-related molecules could contribute to the stabilization of one particular type of secondary structure element at the N-terminus of Mdm2 or MdmX, I believe it would be important for the authors to take those interactions into account.

We agree with the reviewer that in our previous structure (PDB 5MNJ) there is an interaction between MDM2's 430-436 with a symmetry-related UbcH5B molecule and this could contribute to the stabilization of the alpha helix in this particular structure. In the structure of MDM2-MDMX alone (PDB 2VJF), this region adopts a 3-10 helical

configuration and is also involved in crystal packing with a symmetry-related molecule. We have now included Supplementary Figure 4 to illustrate the crystal contacts and stated that this could contribute to the stabilization of the observed structural fold in the heterodimer.

In our human MDM2-419-C-UbcH5B-Ub complex, this region is not involved in crystal contacts (see Supplementary Figure 4a). In both complexes, cat-MDM2-pS429-UbcH5B-Ub and cat-MDM2-S429E-UbcH5B-Ub, a symmetry-related UbcH5B molecule is in close proximity but not involved in direct contacts. For cat MDM2-S429E alone structure, this region is involved in the packing. Nonetheless it still adopts 3-10 helices. For this reason crystal packing had little influence on 3-10 helical configuration in the homodimer.

We have now included few statements in the results section to highlight the symmetry-related interactions.

Minor points:

1. Could the authors indicate the distances for the hydrogen bonds depicted in Figure 4f and 4h.

We have now indicated the distances in the Figure.

2. It would be interesting to directly compare the results obtained with the I440K S429E mutant to the ones obtain with the I440K mutant in Figure 6b,c to show that there are no other effects of S429E in the context of a “dead ligase”.

We have now included the I440K mutant in Figure 6b and 6c showing that it is a dead ligase and S429E has no other effect in this context.

3. In Figure 6d, it appears Mdm2 S429A is less stable than Mdm2 WT in the absence of Etoposide (no band at 90 min for S429A whereas a band is present for WT). Could the authors comment on this?

We thank the reviewer for pointing this out. The actin control was much less at 90 min for S429A mutant as compared to the rest. We have now re-run the blots and quantified.

REVIEWERS' COMMENTS:

Reviewer #1 (Remarks to the Author):

All of my points have been addressed comprehensively, so I recommend the manuscript for publication.

Reviewer #2 (Remarks to the Author):

The authors have very thoroughly addressed all of the reviewer comments and I fully endorse publication of the revised manuscript in Nature Communications.

Reviewer #3 (Remarks to the Author):

The authors addressed all of my concerns by adding new experiments and by clarifying the text. The authors also repeated some of their experiments which boost confidence on the robustness of their findings. They also added quantifications to support their analyses. Overall, the manuscript has improved and I now fully support its publication.

We thank reviewers for their positive comments and recommendation for publication.

REVIEWERS' COMMENTS:

Reviewer #1 (Remarks to the Author):

All of my points have been addressed comprehensively, so I recommend the manuscript for publication.

Reviewer #2 (Remarks to the Author):

The authors have very thoroughly addressed all of the reviewer comments and I fully endorse publication of the revised manuscript in Nature Communications.

Reviewer #3 (Remarks to the Author):

The authors addressed all of my concerns by adding new experiments and by clarifying the text. The authors also repeated some of their experiments which boost confidence on the robustness of their findings. They also added quantifications to support their analyses. Overall, the manuscript has improved and I now fully support its publication.